# Compound Uncertainty Quantification and Aggregation for Reliability Assessment in Industrial Maintenance †

**Alex Grenyer** [1] , **John Ahmet Erkoyuncu** [2,*] , **Sri Addepalli** [2] and **Yifan Zhao** [2]

1   BAE Systems Surface Ships Limited, Warwick House, Farnborough Aerospace Centre, P.O. Box 87, Farnborough GU14 6YU, UK
2   Centre for Digital Engineering and Manufacturing, Cranfield University, Cranfield MK43 0AL, UK
*   Correspondence: j.a.erkoyuncu@cranfield.ac.uk
†   This paper is an extended version of our paper published in A. Grenyer, J.A. Erkoyuncu, S. Addepalli, Y. Zhao, An Uncertainty Quantification and Aggregation Framework for System Performance Assessment in Industrial Maintenance. In Proceedings of the 9th International Conference on Through-life Engineering Service, Cranfield, UK, 3–4 November 2020.

**Abstract:** The mounting increase in the technological complexity of modern engineering systems requires compound uncertainty quantification, from a quantitative and qualitative perspective. This paper presents a Compound Uncertainty Quantification and Aggregation (CUQA) framework to determine compound outputs along with a determination of the greatest uncertainty contribution via global sensitivity analysis. This was validated in two case studies: a bespoke heat exchanger test rig and a simulated turbofan engine. The results demonstrated the effective measurement of compound uncertainty and the individual impact on system reliability. Further work will derive methods to predict uncertainty in-service and the incorporation of the framework with more complex case studies.

**Keywords:** coefficient of variation; global sensitivity analysis; measurement; pedigree; reliability; uncertainty quantification

## 1. Introduction

Uncertainty Quantification (UQ) concerning the maintenance of engineering systems is growing in recognition and rigour as the complexity of such systems surges in the modern world. Complex Engineering Systems (CESs) are comprised of multiple sub-elements including equipment and operators that interact simultaneously and nonlinearly with each other and the environment on multiple levels [1,2]. The consideration of the relationships between elements is vital to understand emergent behaviour to aid decision-making [3]. Complex systems science is a field in itself, the theory of which is widely discussed in the literature [3–6], but is outside the scope of this paper.

The maintenance of complex and non-complex engineering systems exhibits a range of uncertainties from interconnected factors such as quality and the availability of quantitative equipment data, as well as the qualitative influence of operators, expert opinion, experience, and environmental conditions [7]. These uncertainties are represented by varying Probability Distribution Functions (PDFs) and can lead to underestimation or overestimation of maintenance costs, reliability measurement, equipment availability, and delays in maintenance scheduling. Recent research in CESs has explored UQ in micro gear measurements [2], structured surfaces using metrological characteristics [8], correlation uncertainty in gear conformity [9], grey-box energy models for office buildings [10], uncertainty in disassembly line design [11], and others reviewed in various related studies. Many of these approaches only consider quantitative uncertainty given by variability in measured data, rather than the compound aggregation of quantitative and qualitative uncertainties [2,8,10,11]. Methodologies to do this are growing in many areas, but are limited

from an industrial maintenance perspective. This is necessary to obtain a comprehensive understanding of system reliability, as well as the inherent risks and knock-on effects imposed by altering elements within the system. Limited research guiding the aggregation of compound uncertainty sets the focus for this paper.

A six-step framework is presented to quantify and aggregate compound uncertainties to enhance system performance assessment. This will provide maintenance planners with a comprehensive view of the parameters surrounding the above factors to improve decision-making capabilities.

A literature review into uncertainty classification in the context of this paper and techniques to combine quantitative and qualitative uncertainties is given in Section 2. The proposed framework is detailed in Section 3 along with key mathematical formulae, functions, and the assumptions made. Section 4 applies the framework to two case studies: a bespoke heat exchanger test rig comprised of multiple sub-systems, developed at Cranfield University [12], and a simulated dataset for turbofan engine degradation. Individual uncertainties from quantitative and qualitative sources and correlations between them are assessed and aggregated to give a confident indication of system performance. Section 5 discusses the results, strengths, and limitations of the framework along with conclusions and future work in this area.

## 2. Literature Review

### 2.1. Deriving Uncertainty and Risk

The distinction between uncertainty and risk is well documented in the literature, though often blurred in practice. Uncertainty is the degree of—or lack of—knowledge held concerning a given entity, be it measured data, equipment state, environmental conditions, or the accuracy of expert opinion. The resulting risk is the negative impact of uncertainty [13–19].

A confident uncertainty estimate can be positively utilised to aid decision-making. Two key types of uncertainty are described in the *Guide to the Expression of Uncertainty in Measurement* (*GUM*): Type A, sourced from quantitative measured data; and Type B, which considers qualitative technical and expert knowledge or experience, as well as environmental conditions [1,20–25]. The uncertainty of a measured input is given by its standard deviation around the mean value, termed standard uncertainty, set within distribution parameters [1,26]. The distinction between types of uncertainty helps to reduce risk and avoid underestimation or overestimation or of the probability of failure in a system [13,27,28].

Uncertainties can be further derived as epistemic and aleatory. The former emanates from knowledge about the measured entity and can be reduced by obtaining additional data or by refining measurement models. The latter represent variables that can differ each time they are recorded and, therefore, cannot be reduced [10,27–33].

Risk can be defined from numerous perspectives [13,14,34,35]. In a broad sense, it is the probability of the loss or gain of a quantity that holds value. In this context, uncertainty is the lack of knowledge about the degree of risk that exists. It is necessary to distinguish types of uncertainty to reduce risk and avoid underestimation or overestimation or of the probability of failure in a system, which could have significant negative knock-on effects [13,27,28]. Risk assessments in conjunction with uncertainty analysis are highly beneficial in decision-making, but it is necessary to consider other principles, methods, and instruments involved [14]. There is a requirement to look beyond the probabilistic world and embrace subjective and expert opinions.

### 2.2. Combining Quantitative and Qualitative Uncertainty

The *GUM* has been implemented in numerous applications utilising UQ [1,2,9,20,21,23,25,36–39]. The main process defined involves five core stages: (1) identify the measurand; (2) identify uncertainty sources and associated probability distributions; (3) quantify uncertainties (simulation); (4) aggregate uncertainties; (5) report analysis re-

sults. UQ is not always considered a core task, especially considering qualitative factors [40]. Coverage factors are applied to account for purely qualitative estimates or a combination of the two types. While proficient for purely quantitative analysis, the coverage factors have been found to lead to underestimation and cannot be realistically applied in dynamic complex systems [41,42].

UQ in CESs involves the propagation of errors around the sample mean of each parameter via simulation [39]. The three most-common and -validated propagation techniques are Taylor series expansion, Monte Carlo simulation, and Latin Hypercube Sampling (LHS). Taylor series expansion is not suitable for complex nonlinear models, but the derivation of normalised sensitivity coefficients would be beneficial to identify the most-significant parameters [43]. Monte Carlo simulation is widely used, relatively simple, adaptable, and applicable for more complex applications [9,32,44–46]. LHS migrates simple Monte Carlo to assess the convergence of cumulative probability distributions for output variables [6,10,25].

Clarke et al. [25] reviewed the application of these techniques and applied them in a thermodynamic analysis of heat exchanger designs, which highlighted the need to consider both quantitative and qualitative uncertainty and the identification of parameters that pose the greatest influence on uncertainty through Sensitivity Analysis (SA). The approaches used were influenced by Vasquez and Whiting [47]. Similarly, Tatara and Lupia [48] examined heat exchanger performance through temperature measurement uncertainty, with a spotlight on the effect data acquisition methods and measurement devices have on the resulting uncertainty. The heat transfer coefficient was calculated considering the quantified uncertainties.

### 2.2.1. Qualitative Contributions

The consideration of qualitative uncertainty factors can have significant effects on the overall estimate. This is often overlooked in practice or assigned a general bias element considering data acquisition and aleatory factors. The pedigree approach is a widely renowned and verified approach to equate qualitative estimates in line with quantitative data. First proposed by Funtowicz and Ravetz [49], the approach comprises a matrix to score expert knowledge and opinion according to predefined criteria to permit quantitative reliability assessment. This has been applied in a range of fields including oil and gas, meteorology, and genealogy [7,24,38,40,50,51]. It can be applied on its own or through an encompassing approach to standardise combined uncertainty dimensions via five qualifiers: Numeral, Unit, Spread, Assessment, and Pedigree (NUSAP) [24,38,50,52]. The first three terms consider quantitative factors: quantity value, acquisition date, and the random error of the variance of the dataset (addressed by SA and Monte Carlo simulation), respectively.

Ciroth et al. [38] presented a process to improve uncertainty estimation by gauging qualitative uncertainty factors through the pedigree approach for flow data in a multidimensional database. Estimates are attributed by their Geometric Standard Deviation (GSD), where inputs fit the multiplicative lognormal distribution (Equation (1)) [38,53,54]. It is stated that the arithmetic standard deviation used to attribute uncertainty in quantitative data has the disadvantage of relying on the scale (unit) of data in a linear manner [38,53]. Therefore, for the analysis of data from varying sources and measured in different units, uncertainty factors need to be independent of scaling effects. Using the GSD as the uncertainty measure overcomes scale dependency.

$$\sigma_g = \exp\left(\sqrt{\frac{1}{n} \times \sum_{i=1}^{n} ln\left(\frac{x_i}{\bar{x}_g}\right)^2}\right) \tag{1}$$

where: $\sigma_g$ = GSD; $n$ = number of inputs; $x_i$ = dataset; $\bar{x}_g$ = geometric mean of dataset.

To enable aggregation where data sources do not follow a lognormal distribution, GSD ratios are obtained via the Coefficient of Variation (CV) [53,55]. This is a dimensionless

measure of variability defined as the ratio between the standard deviation and the mean [55,56]. Muller et al. [53] provided formulas to apply the CV to various distributions to allow the user to select the most-appropriate types for analysis. This is a key method to aggregate compound uncertainties through different PDFs, given in Table 1, the robustness of which was tested for each parameter PDF using Monte Carlo simulation.

**Table 1.** Probability Distribution Function (PDF) and relative Coefficient of Variation (CV) calculations [38,53].

| Distribution | Parameters | Deterministic Value | PDF | CV Calculation |
|---|---|---|---|---|
| Lognormal | $x$: Input dataset<br>$\mu_g$: Geometric mean<br>$\sigma_g$: Geometric Standard Deviation (GSD) | Median: $\mu_g$ | $f(x, \mu_g, \sigma_g) = \dfrac{\exp\left(-\dfrac{(lnx - ln\mu_g)^2}{2ln^2\sigma_g}\right)}{\sqrt{2\pi}ln\sigma_g}$ | $CV = \sqrt{\exp\left(ln^2\sigma_g\right) - 1}$ |
| Normal | $x$: Input dataset<br>$\mu$: Arithmetic mean<br>$\sigma$: Arithmetic standard deviation | Mean: $\mu$ | $f(x, \mu, \sigma) = \dfrac{\exp\left(-\dfrac{(x-\mu)^2}{2\sigma^2}\right)}{\sigma\sqrt{2\pi}}$ | $CV = \dfrac{\sigma}{\mu}$ |
| Uniform | $x$: Input dataset<br>$a$: Minimum value<br>$b$: Maximum value | Mean: $\frac{a+b}{2}$ | $\begin{cases} f(x, a, b) = \frac{1}{b-a} \ for \ a < x < b \\ otherwise, \ f(x, a, b) = 0 \end{cases}$ | $CV = \dfrac{b-a}{\sqrt{3}(b+a)}$ |
| Triangular | $x$: Input dataset<br>$a$: Minimum value<br>$b$: Maximum value<br>$c$: Most likely value | Most likely value: $c$ | $\begin{cases} f(x, a, b, c) = \frac{2(x-a)}{(b-a)(c-a)} \ for \ a < x < c \\ f(x, a, b, c) = \frac{2(b-x)}{(b-a)(b-c)} \ for \ c < x < b \\ otherwise, \ f(x, a, b, c) = 0 \end{cases}$ | $CV = \dfrac{\sqrt{a^2+b^2+c^2-ab-ac-cb}}{\sqrt{2}(a+b+c)}$ |

Given as a dimensionless measure of variability, the CV can be used as a measure of uncertainty for each input and aggregated to give a representative total. The application of the CV and pedigree aims to convert quality and lack of knowledge into uncertainty figures [53].

### 2.2.2. Correlation and Sensitivity Analysis

Dependencies between input parameters should be accounted for through correlation [1,8,26,39,48,57]. Qualitative uncertainties given by subjective opinion are intuitively correlated in terms of rank rather than linear relationships [6]. Spearman's rank correlation ($\rho$) is, therefore, best suited to consider the correlation between compound uncertainties ($x, y$)—given by Equation (2).

$$\rho_{x,y} = \frac{\sum_{i=1}^{n}\left[\rho(x_i) - \overline{\rho}(x)\right]\left[\rho(y_i) - \overline{\rho}(y)\right]}{\sqrt{\sum_{i=1}^{n}\left[\rho(x_i) - \overline{\rho}(x)\right]^2 \cdot \sum_{i=1}^{n}\left[\rho(y_i) - \overline{\rho}(y)\right]^2}} \tag{2}$$

The significance of positive and negative correlations on the aggregated uncertainty estimate will vary with system complexity, as well as the coefficient value. It is important to remember that correlation is not causation and while two parameters can show a significant correlation, they may not be impacted by one another in practice.

Sensitivity Analysis (SA) identifies parameters whose uncertainty has the greatest relative impact on the system [1,39,58–62]. It gives an illustration of the relationships between different inputs of various PDFs and parameters, as well as those with negligible effects that can be removed. An important tool in uncertainty assessment, design optimisation, and reliability measurement, SA is performed in two ways—local and global. Local Sensitivity Analysis (LSA) explores the change of the quantity of interest around a certain reference point, such as nominal values via partial derivatives. This is the simplest approach, but can prove arduous when applied for a large number of parameters. Global Sensitivity Analysis (GSA) studies the effect over the full range of the input space, typically adopting Monte Carlo techniques.

Groen [63] compared five GSA methods in environmental life cycle assessment: squared standardized regression coefficient, squared Spearman correlation coefficient, key issue analysis, Sobol' indices and random balance design. Spearman correlation coefficients and Sobol' indices were found to give the best overall performance. Generally, the

best method depends on the available data, the uncertainty magnitude, and the goal of the study. Spearman correlation coefficients assume linearity in the system, which is often not the case in practice. Sobol' indices allow for nonlinearity, but assume all parameters to be independent to identify the influence of each input parameter on the output [6,10,58–66]. Correlation coefficients should ideally be established between input parameters [63,67]. Discounting correlation is acceptable when the sensitivity of parameter $x$ is significantly greater than parameter $y$, rendering $\rho_{x,y}$ negligible [68]. Where it is not, discounting correlation can lead to underestimation or overestimation of the resulting uncertainty estimate.

Further from the selection of the best sensitivity approach, Groen [68] compared an analytical and a sampling approach to consider dependant variables in GSA, achievable with small datasets through adjusted regression models devised by Xu and Gertner [67]. Both approaches resulted in relatively equal output variance and sensitivity indices for the applied case study. The sampling approach assumed all inputs to be normally distributed. Knowledge of parameter PDFs, means, standard deviations, and correlations is required prior to sampling—a prerequisite of the model proposed in this paper.

### 2.2.3. Compound Aggregation

The aggregated uncertainty ($U_T$) due to the uncertainty in the quantitative parameters is equal to the Root-Sum-Square (RSS) of those uncertainties ($\sigma$) added to significant correlation coefficients (Equation (3)) [69] ($x_i$ and $y_i$ are parameter $x$ and parameter $y$, where I = 1 for the sum of those parameters; $\sigma_{x_i}$ is the uncertainty for parameter $x_i$). If parameters are independent ($\rho = 0$), the second half of the equation is zero and cancels out. The widely used propagation of error model uses Taylor series expansion to consider local sensitivity coefficients within the aggregation, given by partial derivatives [1,25,39,69]. While suitable for non-complex models, the use of partial derivatives in complex nonlinear models can give a large degree of error and lead to underestimation or overestimation of the uncertainty propagation [25]. This paper, therefore, propagates uncertainties via Monte Carlo simulation and assesses their effect on the output response through GSA, as discussed previously.

$$U_T = \sqrt{\sum_{i=1}^{n} \left( \sigma_{x_i}^2 + \sigma_{y_i}^2 \right) + 2 \left( \rho_{x,y} \sigma_x \sigma_y \right)} \tag{3}$$

To combine quantitative, recorded parameters with qualitative factors, recorded standard deviations are converted to their respective CVs according to their PDF type. The arithmetic mean of symmetric PDFs such as normal and uniform is equal to the mode and, as such, does not change when uncertainty increases [53]. They can, therefore, be aggregated additively by the RSS (Equation (3)). Lognormal distributions are asymmetric; the arithmetic mean will change with increasing or decreasing uncertainty. CVs represented by the lognormal distribution, $CV_{Ln}$, are aggregated multiplicatively by Equation (4) [53]. To combine these with symmetric distributions, a new arithmetic mean needs to be calculated to account for the shifting uncertainty, given by Equation (5) [53]. The proposed approach to aggregate compound uncertainty is discussed in Section 3.

$$CV_{Ln} = \sqrt{\prod_{i=1}^{n} (CV_i^2 + 1) - 1} \tag{4}$$

$$\mu_T CV_T = \mu \sqrt{CV_{Sym}^2 + CV_{Logn}^2} \tag{5}$$

### 2.3. Research Gaps

The *GUM* method is widely adopted for UQ. Along with the propagation of error method, this provides highly confident depictions of purely quantitative uncertainty. However, methods of deriving qualitative uncertainty using the *GUM* have been found to lead to inaccurate depictions [37,70]. Qualitative uncertainties are best accounted for through

the pedigree approach, the criteria for which should be well established when designing the analysis architecture [24,50,51]. The identification of the most-appropriate PDF to represent each input is key to assess its uncertainty [51,71]. This can be achieved visually by comparing fits against a plotted histogram of the data. The representation of uncertainty through the respective CV, as described by Ciroth et al. [38] and Muller et al. [53], enables the quantification and aggregation of compound uncertainties and can be applied to a range of symmetric and asymmetric PDFs. While formulae to denote the inputs of varying PDFs by their respective CVs are defined, a method to aggregate CVs from a mix of symmetric and asymmetric PDFs in a compound manner is unclear. This is necessary to establish compound uncertainty estimates represented by different PDFs with a high degree of confidence.

This compound aggregation can then be used in GSA to calculate sensitivity indices. Correlations should be considered where suitable to avoid underestimation or overestimation in the estimate; however, the majority of applied studies assume input variables to be independent. It is logical to assume there will be significant correlations between quantitative, measured variables and the qualitative influence on how those variables are recorded. Emerging techniques have been proposed to account for dependant variables in SA, with varying success [66–68]. Incorporation with qualitative uncertainties also requires further research at this stage [6,58,61,66]. The risks in ignoring correlation in uncertainty propagation and SA were explored extensively by Groen [63,68]. The role of GSA is to identify variables that have a significant impact on the system, which ties in closely with correlation, though at different stages in the analysis. The consideration of correlation through the sampling GSA approach allows for increased accuracy in the determination of which variables have the most-significant impact on the overall uncertainty and is therefore incorporated in this study [67,68]. The ability to consider PDFs other than normal will further enhance this capability in the aggregation framework.

The research gaps are therefore summarised as:

1. Approaches to quantify and aggregate compound uncertainties represented by different distributions, considering dependencies between them, applicable to increasingly complex engineering systems.
2. Application of GSA to determine the impact of individual uncertainties on the aggregated total, accounting for compound parameters and significant correlation.

## 3. Compound Uncertainty Quantification and Aggregation Framework

Every measurement or estimate is subject to a degree of error, which in turn contributes a level of uncertainty. Quantifying this uncertainty enables a thorough assessment of the scale of risk each component might inflict on the system [1,20]. The level of uncertainty and associated risk can directly or indirectly influence system reliability for maintenance planning, corresponding turnaround times, and system performance.

This paper contributes a holistic assessment of compound uncertainties in dynamic data represented by different distributions with an integrated assessment of correlations and sensitivity. This addresses the research gaps above and was achieved through a six-step modelling approach developed in MATLAB (version 2022b), described below and illustrated in Figure 1.

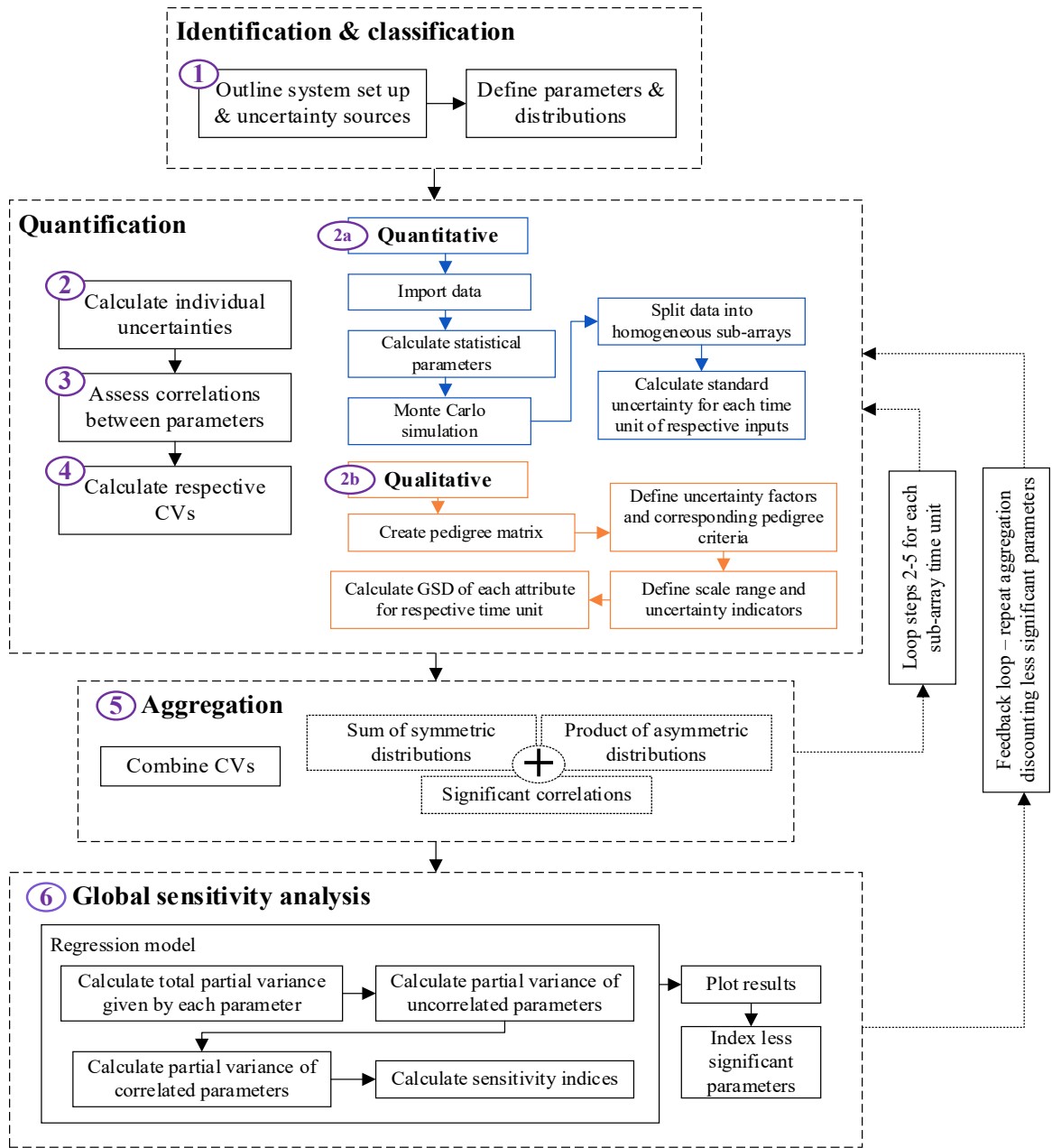

**Figure 1.** CUQA framework overview.

The framework was designed as an extension and amalgamation of existing methodologies from the literature [1,20,33,70,72–74]. An initial version was presented in Grenyer et al. [75]. Here, it is further developed and validated in two distinct case studies to illustrate the framework's flexibility in contextual application, considering key parameter variables identified within a system. While maintenance practices are not discussed directly in the case studies, the compound uncertainty consideration enables greater coherence in system behaviour, reliability, and maintenance requirements. The framework steps were developed from the traditional approach in the *GUM*, extended to consider compound uncertainty and GSA—detailed as follows:

**Step 1: Outline system setup and uncertainty sources**. The inputs were grouped according to their uncertainty type—quantitative or qualitative. This includes all measured data, assumptions made, and environmental predictions. Distribution types were established by "goodness-of-fit" tests. Selected types were indexed for later calculation.

**Step 2: Calculate individual uncertainties**. Statistical parameters were calculated for each input according to their relative distribution via Monte Carlo simulation and the pedigree matrix. These were grouped for each subsystem, for which the standard uncertainties and correlations were determined separately before combining with the whole system, elaborated as follows:

**Step 2a**: Quantitative, recorded data were concatenated in a cell array to allow inputs with a varying number of data points to be considered. Any non-numeric values (including gaps and non-formatted values) were removed. Monte Carlo simulations were run for the relative indexed PDF over a user-defined number of points (default 10,000) or to the size of the largest input parameter. This propagates the input data to a homogeneous array size. It was assumed that individual non-numeric values will not have a significant impact on the statistical parameters of the sub-arrays (such as outliers). In order to consider the uncertainty in the measured values, each dataset ($X_i$) was split into sub-arrays over the recorded time period. The number of rows for each sub-array ($S_i$) can be selected by the user or defined automatically. Possible values for $S_i$ are defined by the number of factors ($N_f$) in the value of the length of the dataset ($\dim(X_i)$). The automatic selection is given by Equation (6). This aimed to select the middle factor, providing enough values to determine the uncertainty at each point while allocating enough sub-arrays to determine the change in uncertainty for the recorded period. Each dataset was then reshaped according to Equation (7), where $S_{i,j}$ is the reshaped sub-array dimension.

$$S_i = \begin{cases} \left[\left(\frac{N_f}{2}\right) + 1\right], N_f < 10 \\ \left[\left(\frac{N_f}{2}\right)\right], \quad N_f \geq 10 \end{cases}, S_i \geq 1 \tag{6}$$

$$X_i \in \mathbb{R}^{\dim(X_i)} \rightarrow X_i \in \mathbb{R}^{S_{i,j}} \tag{7}$$

The arithmetic and geometric mean and deviation were calculated for each sub-array and the full dataset, along with the maximum and minimum values of each input variable. The standard deviation of each time unit was then calculated using the simulated data for each distribution type. For lognormal variables, the mean and standard deviation are given as geometric. Normal and uniform distribution variables are arithmetic [38]. To visualise the data, boxplots for each sub-array were overlaid on the initial dataset. These plots give more detailed information than standard error bars on the change in uncertainty over time with dynamic datasets.

**Step 2b**: Qualitative factors are defined through pedigree criteria. Based on the example implemented by Ciroth [38], the matrix defines uncertainty indicators based on expert judgement. Criteria are defined for each score for each factor, which relates to predefined case-dependent uncertainty measures. The ideal case has a pedigree score of 1, corresponding to minimal uncertainty. Scores of 2-n have progressively higher uncertainties owing to their representative criteria. While there is no limit to the number of scores, typically a maximum of 5–7 was used. The scores for each factor correspond to an uncertainty indicator, the GSD of which was obtained from one or multiple sources (interviews, surveys etc.). These scores will not be fixed over time, and so were pseudo-randomly applied $\pm 1$ of the defined score for each sub-array. If the uncertainty indicators were obtained from a single source, the GSD is given as its square root. If they were obtained from multiple sources, the GSD is given by Equation (1), modelled by the lognormal distribution [38,53,54]. The GSD of less ideal indicators is given as the ratio of the calculated GSD and that of the ideal score for each input, meaning that it is always equal to or greater than 1 [38].

**Step 3: Determine significant correlations between input parameters**. To best determine correlation, the input parameters must be of equal length. For quantitative data, initial recordings prior to Monte Carlo were sampled to the size of the largest parameter length around their respective PDF type. Qualitative parameters were sampled using their uncertainty score as the respective mean and GSD as the standard deviation under a lognormal distribution to achieve a homogeneous sample size. Spearman's correlation coefficient

$\rho$ (Equation (2)) was calculated between each pairwise input parameter, along with their corresponding *p*-values. These were the result of the null hypothesis significance test that determines whether what is observed in the data sample is likely to be true for a wider population. A default significance level ($\alpha$) of 0.05 determines that for *p*-values $< \alpha$, there is a 5% chance that a significant correlation does not exist between those parameters [6,39]. In addition, an ideal limit to define a significant coefficient magnitude is defined by the user as $\rho_{lim}$ and cut-off $\rho_{cutoff}$. If there was not at least one pairwise coefficient for which the absolute value $|\rho| > \rho_{lim}$, the ideal $\rho_{lim}$ was reduced in increments of 0.01 via a "while" loop until the condition was true or the defined $\rho_{cutoff}$ was reached. This enables the user to define the degree of correlation to be included in the aggregation with the assurance that the resulting coefficients are statistically significant. The corresponding input parameters for which the final condition is true were plotted in a correlation matrix and stored for use in Step 5. This matrix provides a visualisation of the correlation magnitude for each parameter with a significantly correlated pair [76].

**Step 4**: **Calculate the CV for each input**. Uncertainties from different data types represented by different PDFs must be considered on an equal scale in order to be aggregated. This was achieved through the CV, explained in Section 2.2, the formulae for which are given in Table 1 [53]. These were calculated within the framework by a sequential algorithm according to the specified input and distribution type. Summary tables were then generated for the compound inputs and correlation, as calculated in Steps 2–3.

**Step 5: Aggregate respective CVs and correlated parameters**. As discussed in Section 2.2.3, symmetric distributions were aggregated additively by the RSS (Equation (3)). Asymmetric distributions, given by lognormal distributions, $CV_{Ln}$, were aggregated multiplicatively by Equation (4) [53]. The framework splits the calculated CVs of quantitative inputs according to the distribution type. The sum of symmetric attributes were added to the product of the lognormal attributes. Comparing this with Equation (3), the aggregated uncertainty is given by $CV_T$ in Equation (8):

$$CV_T = \sqrt{\sum_{i=1}^{n}\left(CV_{sym}^2\right) + \left(\prod_{i=1}^{n}\left(CV_{Ln}^2 + 1\right) - 1\right) + 2\sum_{i=1}^{n}\left(\rho_{x,y}CV_xCV_y\right)} \tag{8}$$

where $(\rho_{x,y}CV_xCV_y)$ is the Spearman correlation coefficient of two parameters *x* and *y* multiplied by their respective CV.

Individual CVs were plotted as bars against the aggregated total, along with a colour bar to visualise the acceptability of relative factors according to predefined scales. The correlation coefficient standardizes the variables and is, therefore, unaffected by changes in scale or units. The formulae allow the aggregated CV of quantitative and qualitative data to be determined as a measure of total uncertainty. Given that CV is the ratio between the standard deviation and the mean, the output follows a normal distribution. The uncertainty can, therefore, be expressed back as the standard deviation via Equation (9).

$$\sigma_T = \sqrt{\sum_{i=1}^{n}\left(\sigma_i\right)^2} = \sqrt{\sum_{i=1}^{n}\left(\mu_i CV_i\right)^2} \tag{9}$$

Steps 2–5 were repeated for each sub-array unit. Summary variables including the individual and aggregated CV were stored and used to calculate the sensitivity indices in Step 6.

**Step 6: Conduct GSA and visualise results**. The relative influence of individual uncertainties on the aggregated total was calculated as the response vector over each sub-array time unit. The sampling approach proposed by Groen [68], influenced by Xu and Gertner [67], was applied to consider the effect of correlated parameters using an adjusted regression model. Results were visualised by a 3D bar plot to show dependant and independent effects against the total, with the same colour scale applied as for Step 5 to illustrate the severity. A feedback loop was then taken back to Step 2 where parameters

with total effects below a defined threshold (default 5%) were discounted. The aggregated uncertainty and sensitivity indices were updated to determine the parameters contributing the greatest impact to the aggregated uncertainty, visualised in the same manner.

## 4. Stepped Implementation and Results of CUQA Framework

### 4.1. Case Study 1: Heat Exchanger Test Rig

The framework was first applied to a bespoke heat exchanger test rig, developed from an initial design by Addepalli et al. [12] with the installation of a motorised pump and digital sensors. The combination of digital and analogue recording, along with qualitative factors discussed below, manifests compound uncertainty in heat exchanger performance. These uncertainties need to be quantified and aggregated to assess their impact on the system, assessed via the heat transfer coefficient [25,48,77]. This was calculated with the resulting uncertainty, derived alongside the CUQA framework as follows:

**Step 1: Outline system setup and uncertainty sources**. The system comprised a hot closed-loop system and a cold open-loop system, illustrated in Figure 2 Component specifications are described in Table 2. Notation relating to Figure 2 is defined in Table 3.

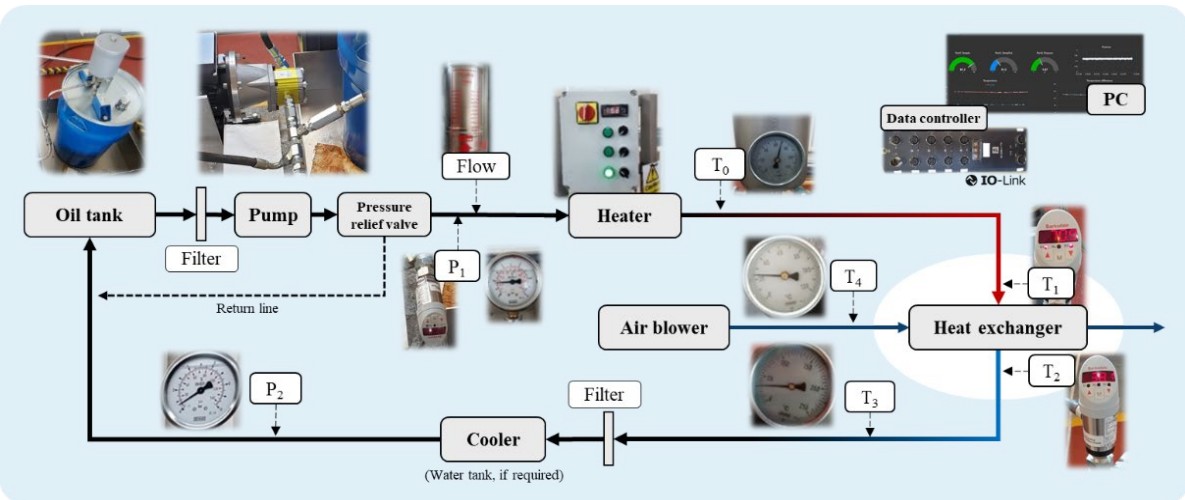

**Figure 2.** Heat exchanger test rig: system design [12].

**Table 2.** Heat exchanger test rig: component specifications of the initial design.

| Component | Specification |
| --- | --- |
| Oil | Aero shell turbine 500 |
| Pump | Vivoil X2P4702EBBA motorised pump |
| Heater | 3 connected units controlled by 3 switches, temp. indicated by probe |
| Heat exchanger | Jaguar oil cooler, plate-fin type |
| Temperature sensors | Barksdale BTS38GVM0050M1 |
| Pressure sensor | Barksdale BPS38GVM0010B |
| IO-Link master | Pepperl + Fuchs ICE2-8IOL-G65L-V1D |

**Table 3.** Heat exchanger test rig: uncertainty sources—measured parameters.

| Parameter | Reading Type | PDF | Reading Interval | Reading Error |
|---|---|---|---|---|
| $T_1$, sensor, hot fluid temp. into HEx (°C) | Digital | Lognormal | 0.1 °C | ±0.1 °C |
| $T_2$, sensor, hot fluid temp. out of HEx (°C) | Digital | Lognormal | 0.1 °C | ±0.1 °C |
| $T_3$, dial, hot fluid temp. out of HEx (°C) | Analogue | Normal | 5 °C | ±2 °C |
| $T_4$, dial, cold fluid temp. (air blower) (°C) | Analogue | Uniform | 2 °C | ±0.5 °C |
| $P_1$, sensor, hot fluid pressure pre-HEx (bar) | Digital | Lognormal | 0.01 bar | ±0.01 bar |
| $P_2$, dial, hot fluid pressure post-HEx (bar) | Analogue | Uniform | 0.5 bar | ±0.3 bar |
| $\dot{V}$, volumetric flow rate of hot fluid (L/min) | Analogue | Uniform | 5 L/min | ±2 L/min |

The experimental setup comprised seven quantitative parameters, summarised in Table 3, along with their corresponding reading interval and error and five qualitative factors: (1) reliability of data, (2) basis of estimate, (3) reading accuracy, (4) environmental conditions, and (5) sample size—each modelled by the lognormal distribution. Oil temperature at the inlet ($T_1$) and outlet ($T_2$) was measured by dual-temperature sensors. A constant flow rate was maintained by a motorised pump. Oil pressure ($P_1$) was regulated by a pressure relief valve, recorded by a dual pressure sensor at the pump outlet. The sensors fed real-time data to the PC controller via IO-Link, logged to a CSV file in 1 s intervals along with a timestamp.

The heat transfer coefficient is given by the heat load $Q$ of the hot ($h$) and cold ($c$) fluid (Equation (10)):

$$Q_h = \dot{m}_h \cdot cp_h \cdot (T_{hIn} - T_{hOut})$$
$$Q_c = \dot{m}_c \cdot cp_c \cdot (T_{cOut} - T_{cIn}) \tag{10}$$

where $\dot{m}$ = mass flow rate, given by the product of the volumetric flow rate $\dot{V}$ and density $\rho$; $cp$ = specific heat capacity; $(T_{In} - T_{Out})$ is the fluid temperature differential in and out of the heat exchanger.

The heat balance error and composite heat load considering associated uncertainty ($U_Q^2$) are given by Equations (11) and (12), respectively, as derived by Tatara and Lupia [48]. Contributing measurement uncertainties and additional qualitative bias in the system were calculated separately using the propagation of error method [39].

While $|HBE| < |U_{HBE}|$ (Equation (13)), the overall heat transfer coefficient can be found and the associated measurement uncertainties were considered valid [48].

$$HBE = \frac{Q_h - Q_c}{Q_h} \cdot 100\% \tag{11}$$

$$Q = \frac{Q_c U_{Q_h}^2 + Q_h U_{Q_c}^2}{U_{Q_h}^2 + U_{Q_c}^2} \tag{12}$$

$$U_{HBE} = 100\% \cdot \frac{Q_h}{Q_c} \sqrt{\left(\frac{U_{m_h}}{m_h}\right)^2 + \left(\frac{U_{ThIn}}{T_{hIn} - T_{hOut}}\right)^2 + \left(\frac{-U_{ThOut}}{T_{hIn} - T_{hOut}}\right)^2 + \left(\frac{U_{m_c}}{m_c}\right)^2 + \left(\frac{U_{TcIn}}{T_{cIn} - T_{cOut}}\right)^2 + \left(\frac{-U_{TcOut}}{T_{cIn} - T_{cOut}}\right)^2} \tag{13}$$

where: $U_x$ = uncertainty in relative parameter

The focus of this study was on the uncertainty in the measured values over time, not the uncertainty of the overall recording period.

The heating system was set to switch off at 80 °C to prevent overheating. However, due to its design, the heater was not able to sustain the temperature at 0.02 °C/min for

10 min, as recommended by Tatara and Lupia [48] to determine the steady-state. While this is unsuitable for thorough thermodynamic assessment of heat transfer efficiency from the heat exchanger, it contributed further qualitative uncertainty to the system, which was reflected in the application of the CUQA framework.

The steady-state region was, therefore, defined by the time of the first and last peak temperature readings at $T_1$. Two cycles were completed, with a total of 85 min recorded; a total of 5590 data points for the three digital parameters. The temperature recorded at $T_1$ had an overall range of 6.8 °C and 1.2 °C at $T_2$ over the recorded period. The pressure, $P_1$ was set at 1.8 bar, following a lognormal distribution with a range of 0.32 bar.

Aside from these readings, all variable measurements were recorded via in-line analogue dials. Many of these dials gave readings on different interval scales and varying measurement accuracy and, therefore, resulted in an increased uncertainty. Additional attributes such as parallax error and ambient temperature further increased the uncertainty in the measurement.

The volumetric flowrate $\dot{V}_h$ of the oil (hot fluid) was held at 5 L/min ($0.83 \times 10^{-3}$ m$^3$/s) with a uniform distribution. A reading error of $\pm 2$ L/min was assigned owing to the scale of the flowmeter. At a maximum temperature of 80 °C, $\rho \approx 0.95$ kg/L (950 kg/m$^3$). Therefore, $\dot{m}_h$ for the hot fluid = 0.08 kg/s. $cp_h$ was given as 1800 J/kg.C. For the air (cold fluid), $\dot{m}_c$ was given as 1.12 kg/s and $cp_c$ as 1005 J/kg.C. Further thermodynamic analysis involving parameters such as oil viscosity and temperature loss through connecting pipes were out of the scope of the framework application. The uncertainty contributed by these factors was factored into the pedigree matrix.

**Step 2a: Calculate quantitative uncertainties**. A summary of the seven quantitative parameters is given in Table 5. Summary statistics from the logged data for $T_1$, $T_2$, and $P_1$ are given by the boxplots in Figure 3. The outliers were values greater than $q_3 + w(q_3 - q_1)$ or less than $q_1 - w(q_3 - q_1)$, where $w$ is the maximum whisker length, 1.5 times the interquartile range, and $q_1$ and $q_3$ are the 25th and 75th quartiles of the respective dataset [78].

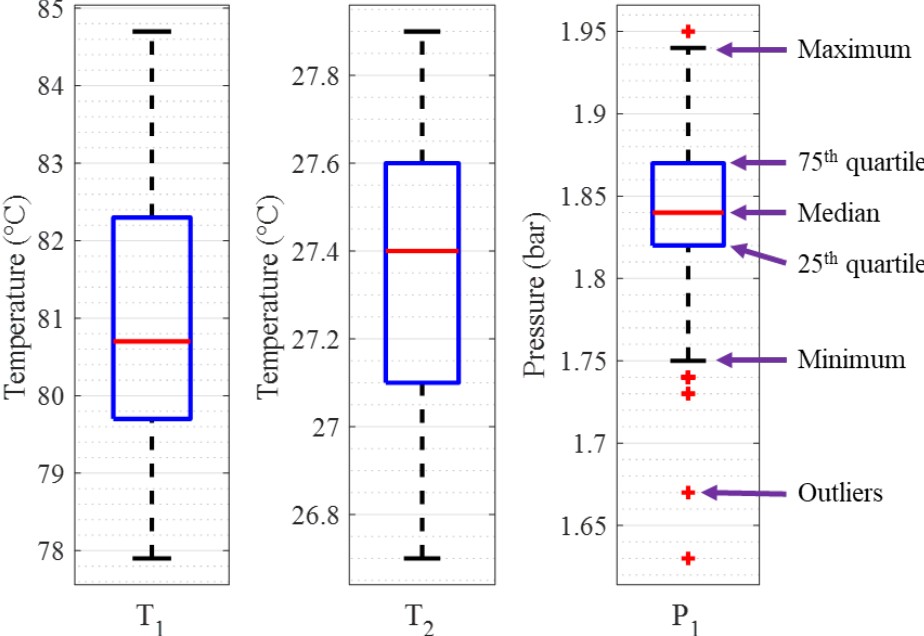

**Figure 3.** Heat exchanger test rig: boxplots for $T_1$, $T_2$, and $P_1$.

The three digitally recorded parameters were split into 65 homogeneous sub-arrays over the 5590 data points. The overlaid boxplots are shown in Figure 4 (coloured as for Figure 3), plotted over the time series of the logged data. Owing to the multimodal shape of the data, the sub-array standard deviation for $T_1$ was low to negligible at the peaks and troughs and high for temperature increases or decreases. The temperature at $T_2$ was more

constant with respect to $T_1$, showing a step change over time owing to the heat transfer coefficient of the heat exchanger.

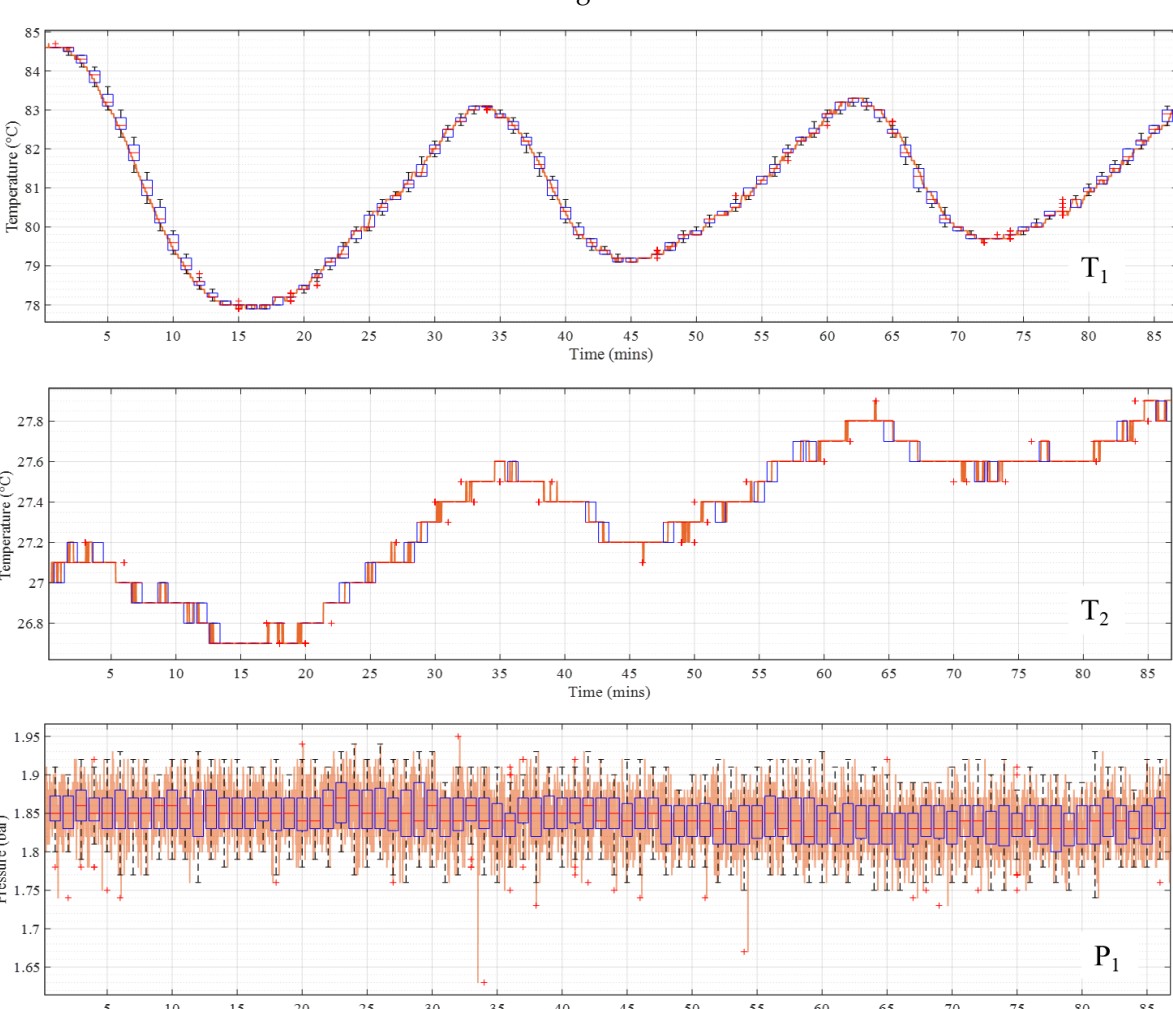

**Figure 4.** Heat exchanger test rig: sub-array boxplots over time series data.

The greater the sub-array size ($S_i$), the greater the uncertainty in the measurement. This is illustrated in Appendix A (Figure A1) for all possible factors (a), with a focus on $S_i$ values of 0–130 and the automatically selected value, 86, highlighted (b). This procedure enables a mean uncertainty estimate to be obtained where the recorded data are not able to meet the criterion for steady-state readings. As $S_i$ increased, the number of sub-arrays decreased, resulting in greater uncertainty. This was considered by the basis of estimate factor in the pedigree matrix.

The four remaining quantitative parameters were acquired by analogue dials with varying reading intervals (Table 3). These were taken every 30 min over the recording period, resulting in limited data in comparison to the automated recording. Using Monte Carlo simulation, the readings were propagated to match the array size of the three digital parameters according to their statistical range and rounded to their corresponding reading intervals.

**Step 2b: Calculate qualitative uncertainties**. The five qualitative factors were scored by the defined pedigree criteria detailed in Table 4. These were based on adjusted examples from the literature to apply to the case study [7,24,52]. Uncertainty indicators for each factor for increasing pedigree scores corresponding to the criteria are illustrated in Figure 5. For this case study, the uncertainty indicators were obtained from a single source (the authors opinion), applied to the full dataset. Their GSDs are, therefore, given as the square root of

the uncertainty indicator. These scores will not remain fixed over time and were, therefore, pseudo-randomly applied ±1 of the defined score circled in Figure 5 for each sub-array.

**Table 4.** Heat exchanger test rig: pedigree criteria.

| Score | 1 | 2 | 3 | 4 | 5 |
|---|---|---|---|---|---|
| Reliability of data | Data are <2 months old and/or recorded by fully calibrated sensor or fully qualified person | Data are <6 months old and/or recorded by fully qualified person, but sensor requires recalibration | Data are <12 months old and/or recorded by experienced person, but sensor requires recalibration | Data are >12 months old and/or recorded by experienced person, sensor accuracy unknown | Age or source of data unknown or >12 months old |
| Basis of estimate | Best-possible data, use of historical field data, validated tools, and independently verified data, given by fully qualified person | Smaller sample of historic data, parametric estimates, internally verified data, some experience in the area | Limited available data, unverified, inexperienced opinions | Incomplete data, small sample, educated guesses, indirect approximate rule of thumb estimate | No experience in the data |
| Reading accuracy | Measurements taken using fully calibrated and accurate equipment: ±0.01 °C, ±0.1 bar | Measurements taken using recently calibrated, but less accurate equipment: ±0.1 °C, ±0.5 bar | Measurements taken using recently calibrated, but less accurate equipment: >±1 °C, >±2 bar | Measurements taken using accurate equipment that may need recalibrating | Measurements taken using un-calibrated and inaccurate equipment |
| Environmental conditions | Data recorded under specific consistent conditions or a specified range of conditions from area under study | Data recorded in generally consistent conditions with fluctuations specified | Data recorded in generally consistent conditions, changes not specified | Data recorded in a range of unspecified conditions | Data from unknown or distinctly different areas |
| Mean sample size | >1000 | >100 | >50 | <50 | Unknown |

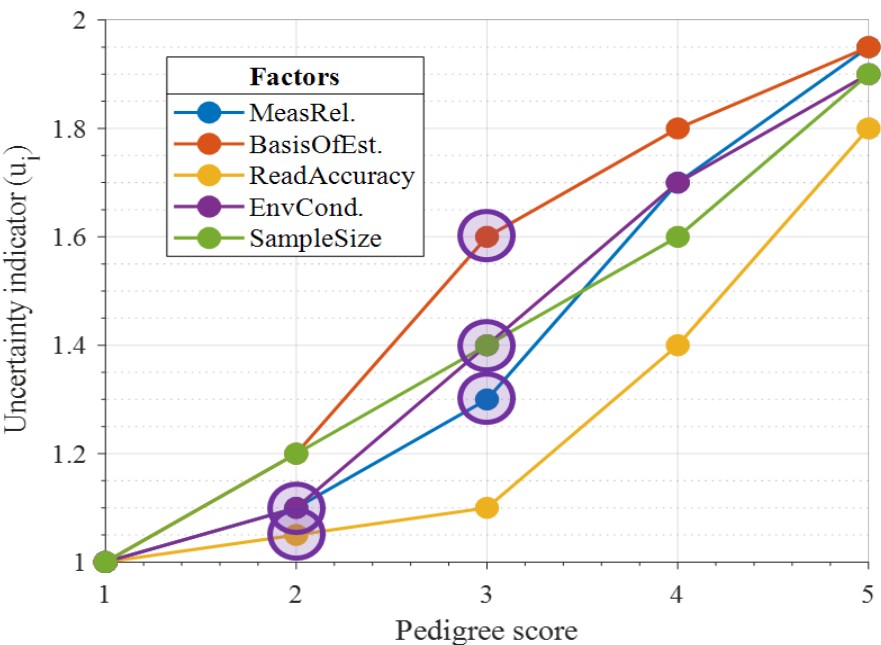

**Figure 5.** Heat exchanger test rig: uncertainty indicators for increasing pedigree scores.

The resulting CV calculated in Step 4 was significantly greater than that of the log-normal recorded data. This was most likely due to the small number of data points in the sub-arrays. To give a closer comparison of the uncertainty, the pedigree factors were rescaled by Equation (14). The following results up to Step 6 illustrate an example for the first sub-array time unit.

$$U_i\_scaled = \frac{(U_i - 1)}{10} + 1 \qquad (14)$$

where $U_i$ = uncertainty indicator.

**Step 3: Assess correlations between parameters**. The ideal limit of $\rho$ was set to 0.5, with a cut-off at 0.2. Naturally, significant positive correlation was identified between $T_1$ and $T_2$, highlighted in red (Figure 6). The negative correlation to $P_1$ reflects the pressure

drop due to oil viscosity with increasing temperature. This shows the effectiveness of selecting the desired $\rho$ limit to remove minor correlations from the analysis.

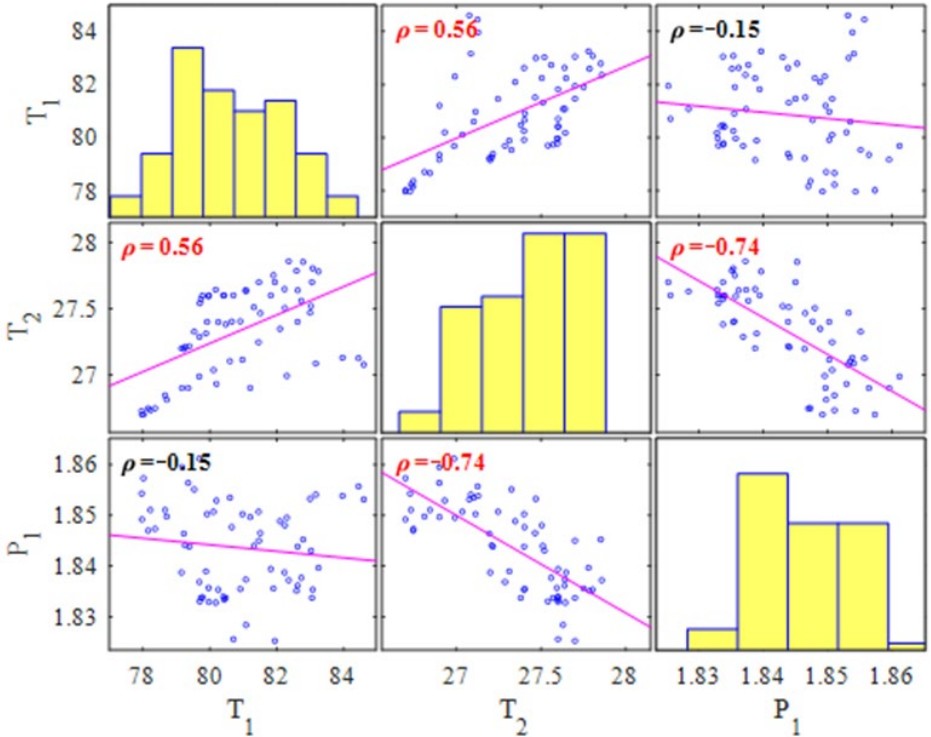

**Figure 6.** Heat exchanger test rig: significant correlations for which $|\rho| \geq 0.5$.

**Step 4: Calculate respective CVs**. The summary tables with the calculated CV for each input are given in Tables 5 and 6 for the quantitative and qualitative factors, respectively. Uniformly distributed parameters had a negligible deviation and, therefore, a CV of zero and did not contribute to the aggregated uncertainty total.

**Table 5.** Heat exchanger test rig: recorded data and calculated parameters.

| Parameter | Reading Interval | Reading Error | Distribution | Mean | Standard Deviation | Min | Max | CV |
|---|---|---|---|---|---|---|---|---|
| $T_1$ (°C) | 0.1 °C | ± 0.1 °C | Lognormal | 80.8654 | 1.0209 | 77.9628 | 84.6012 | 0.0207 |
| $T_2$ (°C) | 0.1 °C | ± 0.1 °C | Normal | 27.3305 | 0.3351 | 26.7 | 27.8581 | 0.0123 |
| $T_3$ (°C) | 5.0 °C | ±2.0 °C | Lognormal | 24.6 | 2.881 | 20 | 28 | 0.1171 |
| $T_4$ (°C) | 2.0 °C | ±0.5 °C | Uniform | 21.6 | 0.8944 | 20 | 22 | 0 |
| $P_1$ (bar) | 0.5 bar | ±1.0 bar | Normal | 1.8436 | 0.0088 | 1.8252 | 1.8612 | 0.0048 |
| $P_2$ (bar) | 0.5 bar | ±0.3 bar | Uniform | 0.9 | 0.2236 | 0.5 | 1 | 0 |
| Flow (L/min) | 5 L/min | ±2 L/min | Uniform | 4.9575 | 0.0253 | 4.9343 | 4.9865 | 0 |

**Table 6.** Heat exchanger test rig: pedigree factors with relating GSD and CV.

| Factor | Distribution | Pedigree Score | Uncertainty Indicator | GSD | CV |
|---|---|---|---|---|---|
| Meas. Relbl. | Lognormal | 2 | 1.1 | 1.0488 | 0.0477 |
| Basis of Est. | Lognormal | 2 | 1.2 | 1.0954 | 0.0914 |
| Read Accuracy | Lognormal | 1 | 1 | 1 | 0 |
| Envir. Cond. | Lognormal | 2 | 1.1 | 1.0488 | 0.0477 |
| Sample Size | Lognormal | 3 | 1.4 | 1.1832 | 0.1694 |

**Step 5**: **Combine CVs**. The combined CV of each PDF was calculated by Equation (8) and summarised in Table 7, aggregated for symmetric and asymmetric distributions and total CV with correlation between $T_1$ and $T_2$—given in the table as $2\left(\rho_{T_1,T_2} \cdot CV_{T_1} \cdot CV_{T_2}\right)$.

**Table 7.** Heat exchanger test rig: CV aggregation results.

| PDF | CV Comb. | CV Agg. | Corr. | CV$_T$ |
|---|---|---|---|---|
| Ln recorded | 0.0207 | 0.2256 | 0.0001 | 0.2593 |
| Ln pedigree | 0.2050 | | 0.0011 | |
| Norm. recorded | 0.1179 | 0.1179 | | |
| Uni. Recorded | 0 | | | |

The visualisation in Figure 7 illustrates the relative CV of each quantitative (blue), qualitative (orange), and correlated (purple) input against the aggregated total (cream) for 1 of the 86 sub-array time units. When calculated for only the quantitative parameters, the aggregated CV fell to 0.1293, a percentage decrease of 50.1%. This illustrates the significance of accounting for qualitative factors alongside quantitative parameters—providing a holistic view of factors that manifest uncertainty in the system. While the depiction of these factors is subjective, the compound consideration reduced the risk of underestimating the aggregated uncertainty, which can occur if only accounting for quantitative parameters [38].

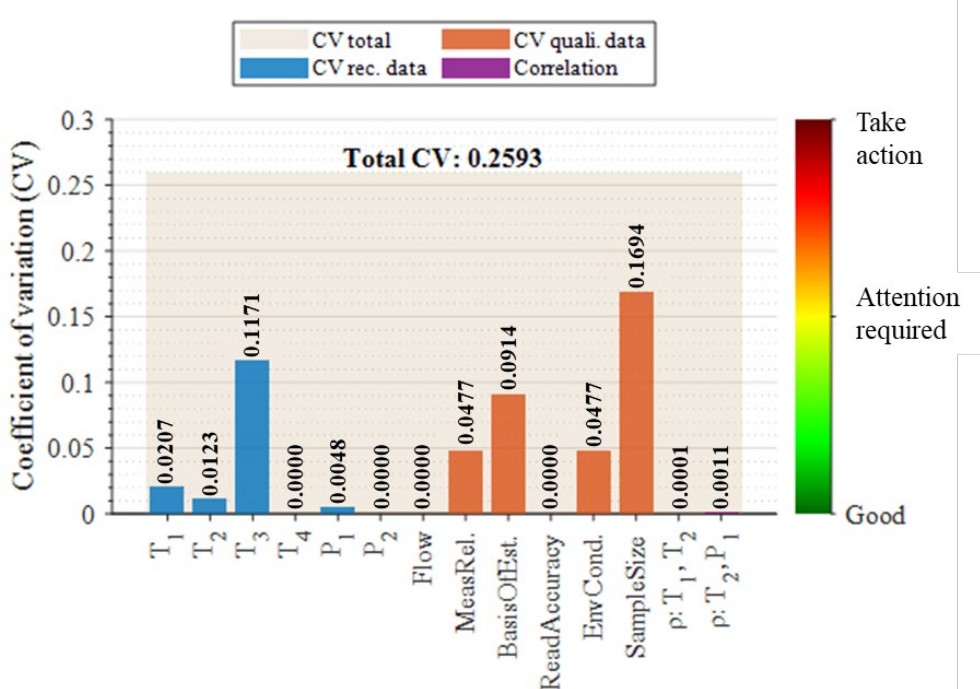

**Figure 7.** Heat exchanger test rig: aggregated total CV against individual factors for one time unit.

Individual uncertainties were then expressed as variances by the square of Equation (9) to feed into Step 6. The change in individual and aggregated CV over all time units for $S_i = 65$ (86 sub-arrays) is given in Figure 8a and compared with $S_i = 215$ (26 sub-arrays) in Figure 8b. This demonstrates the effect of sub-array size on the resulting uncertainty estimate.

Calculating the heat load parameters from Equations (10)–(13) gave [48]: $Q_h = 366.2$ MW, $U_{Qh} = 16.31$ MW, $Q_c = 4.52$ kW, $U_{Qc} = 97.56$ W, and a resulting $Q = 4.52$ kW. The heat balance error (HBE) = 99.98%, and the composite load uncertainty $U_{HBE} = 311\%$. This passed the validity test given by as $|HBE| < |U_{HBE}|$, indicating that the measurements were valid.

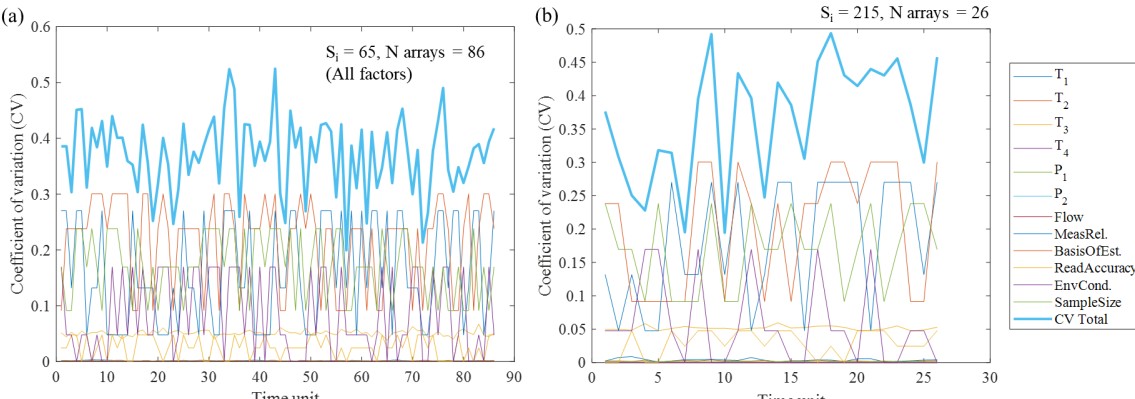

**Figure 8.** Heat exchanger test rig: aggregated total CV against individual factors over all time units for $S_i = 65$ (**a**) and $S_i = 215$ (**b**).

**Step 6: GSA and visualisation**. The relative influence of individual uncertainties on the aggregated total is plotted in Figure 9a. The uncertainty in $T_3$, the oil temperature after being cooled by the heat exchanger, had an overwhelmingly greater effect (76%) on the aggregated uncertainty than any other parameter. This was due to the large error margin of $\pm 2$ °C given by the reading interval on the dial. If $T_3$ was discounted, along with parameters with an impact below 5% (uniformly distributed), the basis of the estimate was deemed to have the greatest effect at 56% (Figure 9b). The influence of T1 and T2 was minimal due to the comparatively equal deviation for each sub-array time unit. The qualitative factors saw greater variability owning to the pseudorandom score allocation (Figure 5).

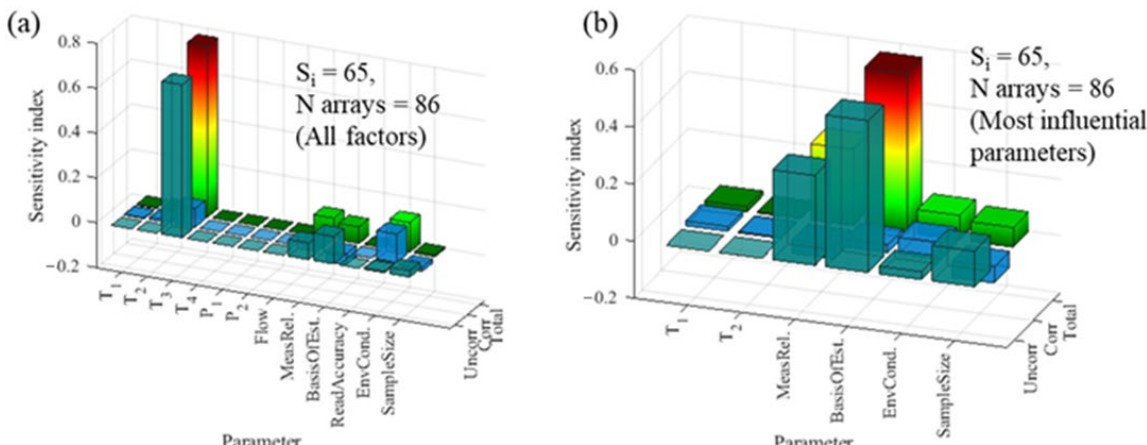

**Figure 9.** Heat exchanger test rig: GSA results of individual to aggregated uncertainty for all factors (**a**) and most-influential parameters (**b**) for test example.

Altering about the pedigree score allocation of the qualitative factors impacted the degree of uncertainty each factor would contribute to the aggregated total, according to the defined uncertainty indicators in Figure 5. Applying higher pedigree scores will apply a higher representative level of uncertainty. The difference between one uncertainty indicator to another will influence the respective factor's sensitivity index owing to the pseudorandom score allocation. Increasing the degree of allocation (e.g., from $\pm 1$ to $\pm 2$) will also influence the respective sensitivity indices, though this was not deemed necessary in this study for the score range of 1–5. While the uncertainty indicator scores were subjective, they were expected to increase linearly or exponentially. Therefore, lower scores would have less influence on the aggregated total.

### 4.2. Case Study 2: Turbofan Engine Degradation

The framework was applied to a turbofan engine degradation dataset from the Commercial Modular Aero-Propulsion System Simulation (C-MAPSS) tool, developed by NASA [79,80]. This publicly available dataset has been widely applied in Prognostics and Health Management (PHM) [80–82]. The C-MAPSS data consist of four datasets simulated under different operating conditions. The FD001 training dataset, simulating the degradation of the High-Pressure Compressor (HPC), was applied to the CUQA framework to analyse the aggregated uncertainty in the measurements over time:

**Step 1: Outline system setup and uncertainty sources**. The FD001 dataset consisted of 21 sensors measuring temperature, pressure, and speed for 100 engine units, each with a random start time and normal operating level, running to failure. For this study, one engine unit was selected with 192 cycles to failure. The system design is illustrated in Figure 10.

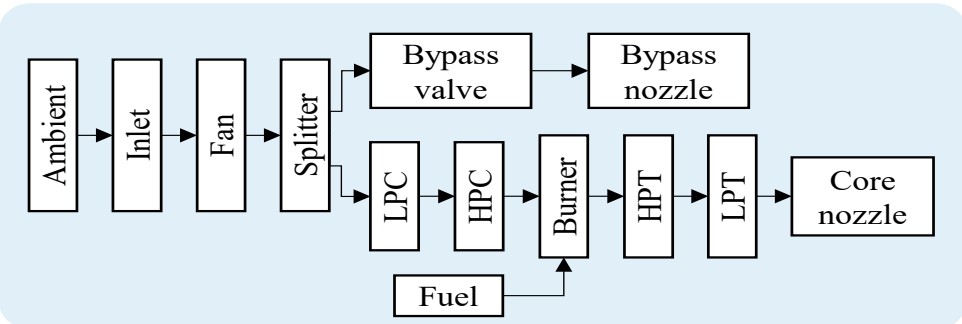

**Figure 10.** C-MAPSS turbofan engine: system design as simulated in C-MAPSS [79].

Previous work using this dataset focused on Remaining Useful Life (RUL) prediction [81,82]. In these studies, sensor data were divided into three categories according to the data trend; ascending, descending, and irregular/constant. Data that did not exhibit an ascending or descending trend over time (uniform) are not viable for RUL prediction and were, therefore, discounted from the dataset. The previous case study showed that constant, uniform parameters did not contribute to the uncertainty. Therefore, the same approach was applied here. A description of the 14 included sensors is given in Table 8.

**Table 8.** C-MAPSS turbofan engine: detailed description of sensors [79].

| Sensor Number | Notation | Description | Unit |
|---|---|---|---|
| 2 | T24 | Total temperature at LPC inlet | °R (Rankine scale) |
| 3 | T30 | Total temperature at HPC inlet | °R |
| 4 | T50 | Total temperature at LPT inlet | °R |
| 7 | P30 | Total pressure at HPC outlet | psi abs. (pounds per square inch, absolute) |
| 8 | Nf | Physical fan speed | rpm (revolutions per minute) |
| 9 | Nc | Physical core speed | rpm |
| 11 | Ps30 | Static pressure at HPC outlet | psi abs. |
| 12 | Phi | Ratio of fuel flow to Ps30 | psi |
| 13 | NRf | Corrected fan speed | rpm |
| 14 | NRc | Corrected core speed | rpm |
| 15 | BPR | Bypass ratio | – |
| 17 | htBleed | Bleed enthalpy | – |
| 20 | W31 | HPT coolant bleed | lbm/s (pound mass per second) |
| 21 | W32 | LPT coolant bleed | lbm/s |

**Step 2a: Calculate quantitative uncertainties**. The sensor data were indexed and divided into 16 sub-arrays consisting of 12 rows by Equation (6). The mean and deviation of each array were calculated up to the point of failure. This is illustrated for 4 of the

14 inputs in Figure A2. A comparison of the sub-array size to the mean deviation is given in Figure A3. Other than for the derivation of pedigree factors in Step 2b, the illustrated results up to Step 6 give an example for the first sub-array unit. A summary of the quantitative sensor data for this example is given in Table 10.

**Step 2b: Calculate qualitative uncertainties**. Random noise models of mixed distributions were used in the composition of the C-MAPSS dataset to propagate associated qualitative factors with a mix of distributions to give realistic results [79,81]. This was given as a combination of three core factors applied to all sensors: manufacturing and assembly variations (resulting in varying degrees of initial wear), process noise (factors not taken into account in modelling), and measurement noise. More in-depth factors concerning maintenance between flights and environmental operating conditions could be considered in practice. For this study, they were incorporated in the three core factors for the simulated data, scored against the pedigree criteria detailed in Table 9 [79]. Uncertainty indicators for each factor are illustrated in Figure 11, with the GSD given as the square root of the uncertainty indicator. As for the previous study, the scores were pseudo-randomly applied ±1 of the defined score circled in Figure 11 for each sub-array, scaled by Equation (14).

**Table 9.** C-MAPSS turbofan engine: pedigree criteria.

| Score | 1 | 2 | 3 | 4 | 5 |
|---|---|---|---|---|---|
| Manufacturing and assembly variations | Negligible range of initial wear on components, not contributing to engine efficiency | Minimal range in initial wear on engine components | Notable range in initial wear on engine components, occasional reduction in engine efficiency | Notable range in initial wear on engine components, regular reduction in engine efficiency | High range in initial wear on engine components, high variance in engine efficiency |
| Process noise | Negligible trend in degradation trajectory, no noise | Minor trend in degradation trajectory, minimal noise | Minor trend in degradation trajectory, manageable noise | Significant trend in degradation trajectory, variable noise | Highly contaminated degradation trajectory |
| Measurement noise | Negligible sensor noise, no impact | Minimal sensor noise, minor impact, predictable trend | Notable random complex sensor noise, measurable impact | Significant random complex sensor noise, inaccurate impact measurement | High random complex sensor noise, tangible point estimate unobtainable |

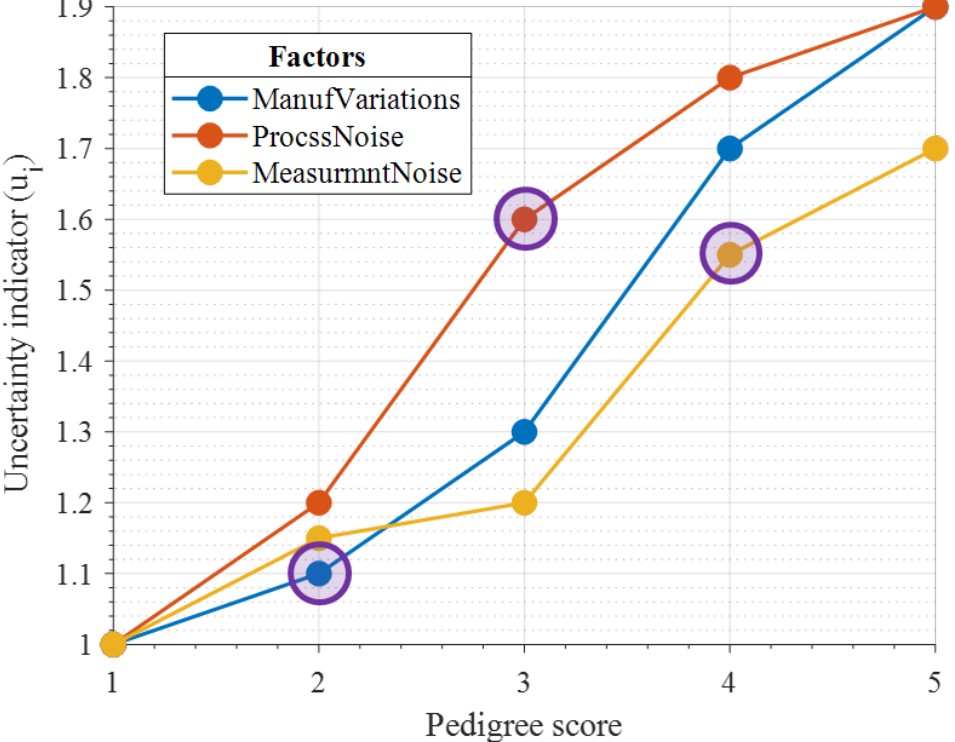

**Figure 11.** C-MAPSS turbofan engine: uncertainty indicators for increasing pedigree scores.

**Step 3: Assess correlations between parameters**. Each sub-array consisted of 12 data points. The ideal limit of $\rho$, therefore, needed to be set to a high level of 0.8, with a cut-off at 0.6. No significant correlations were present above 0.8, so the value was reduced incrementally to 0.78, for which a significant correlation was detected between the pressure at the HPC outlet and turbine core speed (Figure 12a). While it is logical to expect a positive relationship between these parameters, notable in the plot, it was not maintained through the other 15 sub-arrays. This does not mean the relationship was not present, but that other dependencies were more prevalent below the limit of 0.8. When run for all data points, a positive trend was identified between the physical and corrected core speed of the engine (Figure 12b).

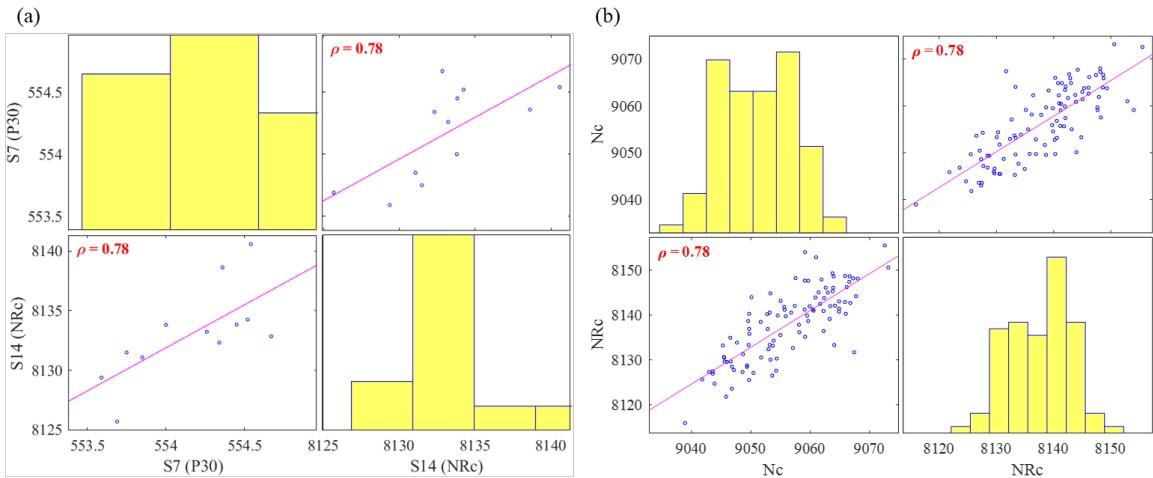

**Figure 12.** C-MAPSS turbofan engine: significant correlations for which $|\rho| \geq 0.6$ for HPC outlet only (**a**) and all data points (**b**).

**Step 4: Calculate respective CVs**. Summary tables with the calculated CV for each input are given in Tables 10 and 11 for the quantitative and qualitative factors, respectively. The majority of factors here were lognormally distributed by the goodness-of-fit tests.

**Table 10.** C-MAPSS turbofan engine: recorded data and calculated parameters.

| Parameter | Distribution | Mean | Deviation | Min | Max | CV |
|---|---|---|---|---|---|---|
| S2 (T24) | Lognormal | 642.20 | 1.0004 | 641.71 | 642.56 | 0.0004 |
| S3 (T30) | Lognormal | 1586.85 | 1.0026 | 1581.75 | 1592.32 | 0.0026 |
| S4 (T50) | Lognormal | 1400.76 | 1.0021 | 1394.80 | 1406.22 | 0.0021 |
| S7 (P30) | Lognormal | 554.17 | 1.0007 | 553.59 | 554.67 | 0.0007 |
| S8 (Nf) | Lognormal | 2388.05 | 1.0000 | 2388.00 | 2388.11 | 0.0000 |
| S9 (Nc) | Normal | 9049.55 | 4.9243 | 9040.80 | 9059.13 | 0.0005 |
| S11 (Ps30) | Lognormal | 47.25 | 1.0029 | 47.03 | 47.49 | 0.0029 |
| S12 (Phi) | Lognormal | 522.05 | 1.0008 | 521.40 | 522.86 | 0.0008 |
| S13 (NRf) | Lognormal | 2388.04 | 1.0000 | 2388.01 | 2388.08 | 0.0000 |
| S14 (NRc) | Lognormal | 8133.09 | 1.0005 | 8125.69 | 8140.58 | 0.0005 |
| S15 (BPR) | Lognormal | 8.41 | 1.0027 | 8.37 | 8.43 | 0.0027 |
| S17 (htBleed) | Lognormal | 391.75 | 1.0022 | 390.00 | 393.00 | 0.0022 |
| S20 (W31) | Lognormal | 38.99 | 1.0018 | 38.88 | 39.10 | 0.0018 |
| S21 (W32) | Lognormal | 23.40 | 1.0021 | 23.31 | 23.48 | 0.0021 |

**Table 11.** C-MAPSS turbofan engine: pedigree factors with related GSD and CV.

| Factor | Distribution | Pedigree Score | Uncertainty Indicator | GSD | CV |
|---|---|---|---|---|---|
| ManufVariations | Lognormal | 2 | 1.01 | 1.0488 | 0.0477 |
| ProcssNoise | Lognormal | 3 | 1.06 | 1.0954 | 0.0914 |
| MeasurmntNoise | Lognormal | 4 | 1.05 | 1.3038 | 0.2701 |

**Step 5**: **Combine CVs**. The combined CV is summarised in Table 12, aggregated for symmetric and asymmetric distributions and the total CV with correlation. The visualisation in Figure 13 illustrates the relative CV of each input against the aggregated total for the example time unit.

**Table 12.** C-MAPSS turbofan engine: CV aggregation results.

| PDF | CV Comb. | CV Agg. | Corr. | $CV_T$ |
|---|---|---|---|---|
| Ln recorded | 0.00639 | 0.29049 | $2.7168 \times 10^{-7}$ | 0.2905 |
| Ln pedigree | 0.29042 | | | |
| Norm. recorded | 0.000544 | 0.000544 | | |
| Uni. recorded | 0 | | | |

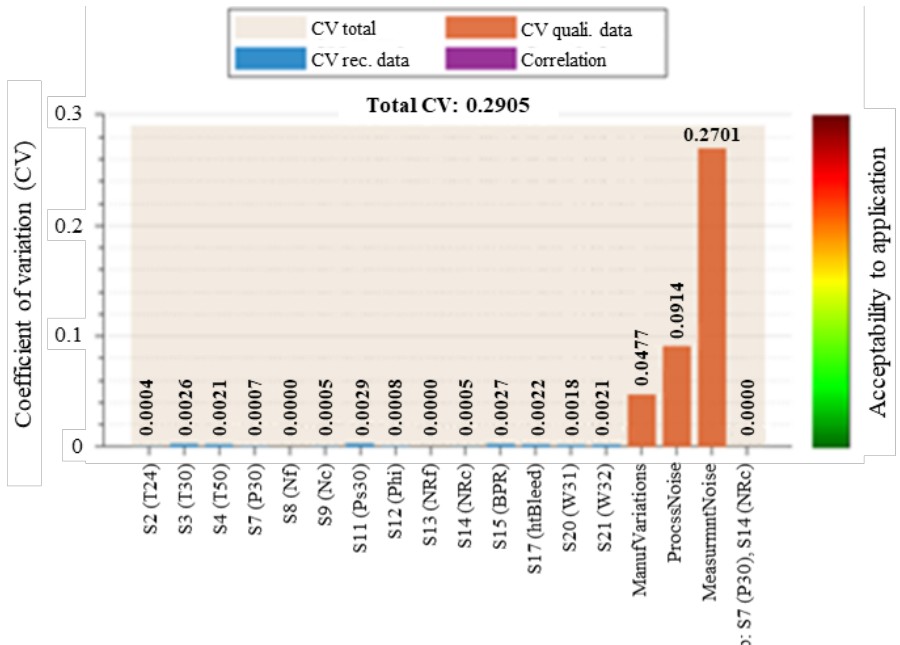

**Figure 13.** C-MAPSS turbofan engine: aggregated total CV against individual factors for one time unit.

For the example time unit, the measured data had minimal uncertainty compared to the qualitative factors. Discounting the qualitative factors here resulted in a 97.8% decrease in the aggregated CV from 0.2905 to 0.0065. The minimal quantitative uncertainty was due to the spread of the 12 data points in the sub-array. Increasing the number of data points increased the mean uncertainty depending on the variability in the dataset, but reduced the number of sub-arrays (Figure A3). The change in individual and aggregated CV over all time units for $S_i = 12$ (16 sub-arrays) is given in Figure 14a and compared with $S_i = 3$ (64 sub-arrays) in Figure 14b.

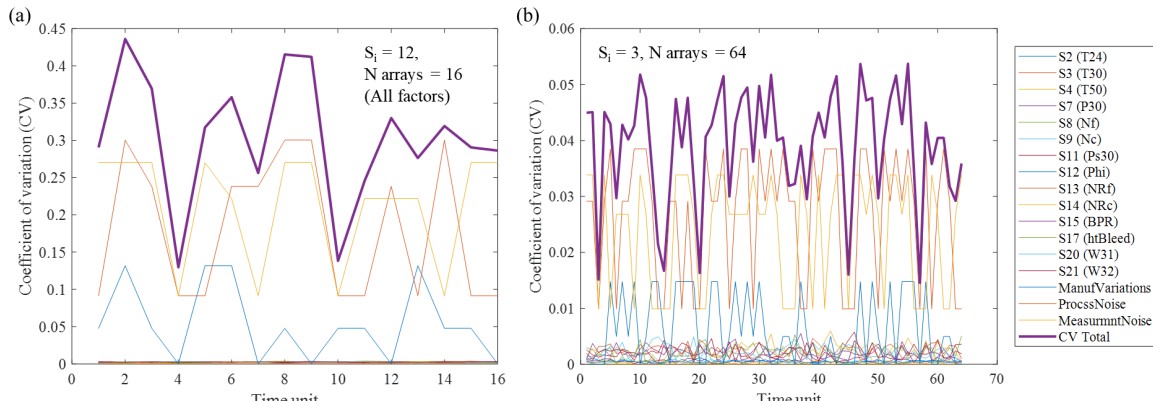

**Figure 14.** C-MAPSS turbofan engine: aggregated total CV against individual factors over all time units for $S_i = 12$ (**a**) and $S_i = 3$ (**b**).

**Step 6: GSA and visualisation**. The relative influence of individual uncertainties on the aggregated total is plotted in Figure 15a and resulted after discounting the less influential parameters factors in Figure 15b. The quantitative parameters had a greater influence than the qualitative factors, despite them having a lower CV for all sub-array units. The most-influential parameter uncertainty was T50 (temperature at LPT inlet) at 37%. Discounting parameters with an impact <5% resulted in Nc (turbine core speed) having a dominating influence, while T50 dropped to 9%. This was again due to the variation in the data points of each sub-array. As for Case Study 1, the difference between one uncertainty indicator to another, defined in Figure 11, will influence the respective factor's sensitivity index owing to the pseudorandom score allocation.

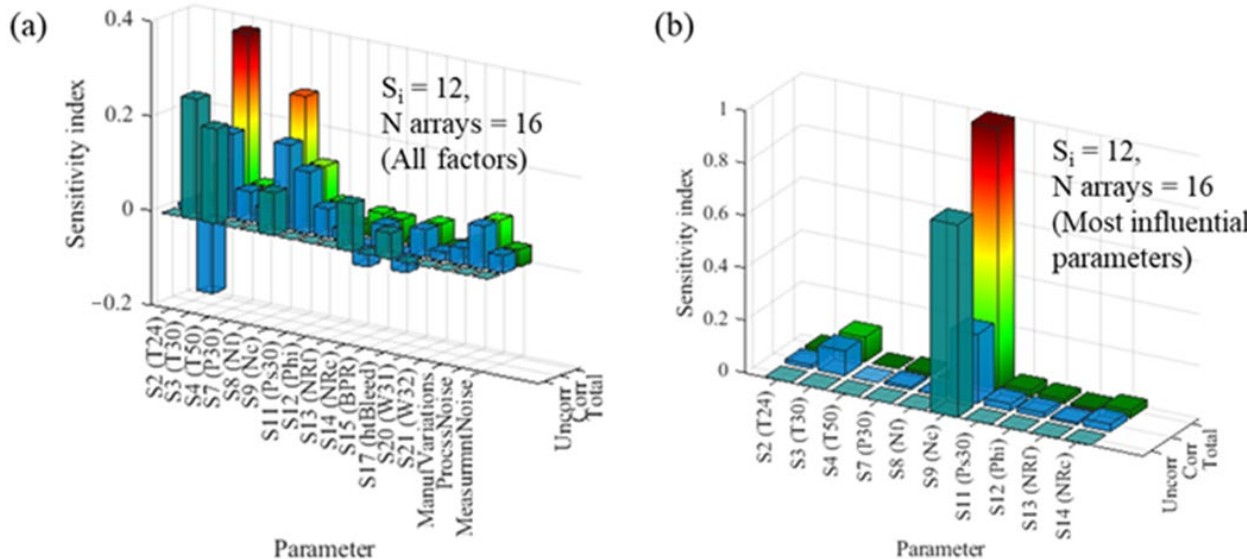

**Figure 15.** C-MAPSS turbofan engine: GSA results for all factors (**a**) and most-influential parameters (**b**) for test example.

## 5. Discussion and Conclusions

The CUQA framework presented in this paper was designed to enhance system reliability measurement in a manner applicable to complex and non-complex engineering systems through quantification and aggregation of compound uncertainties. These develop as a result of the recording methods and assumptions made about the system and were modelled by different distribution types. The framework builds on existing literature to aggregate compound uncertainty considering dependant variables in the analysis, as well as the identification of the greatest contributing factors through the GSA. The benefits

of this framework include enhancements to performance assessment and corresponding maintenance planning for complex and non-complex engineering systems and respective subsystems.

The framework was first applied to a bespoke heat exchanger test rig, which contributed a range of uncertainties that impacted measurement quality and accuracy. Three distributions were considered: lognormal, normal, and uniform. All qualitative factors were lognormal [38]. The measured parameters were deemed valid, though the true steady-state was not obtainable owing to the heating system [48]. The second case study implemented a simulated engine degradation dataset [79]. The majority of the selected sensors exhibited a lognormal distribution up to failure. The following paragraphs critique the effectiveness of the framework through the results of the two case studies, concluding with a summary of the contributions and recommendations for future work.

The CUQA framework is capable of assessing uncertainty for nonhomogeneous input data. The user can view and select the best-suited distribution for each input via "goodness-of-fit" tests. While effective for a small number of inputs, an automated method would prove more efficient for more complex systems. Monte Carlo simulation was used in Step 2a to give a homogeneous array size, enabling level consideration of each input. Monte Carlo was selected due to its flexibility with multiple distributions [6]. The inherently random nature of the simulation, though within the respective distribution parameters, caused different results each time the experiment was run, which may impact the accuracy of the parameter values. Other techniques such as Latin Hypercube Sampling (LHS) and Taylor series expansion may provide samples tighter to the respective mean, but do not show the same flexibility as Monte Carlo for multiple distribution types.

Splitting the input data into sub-arrays enabled uncertainty in the measured values to be determined over time. The greater the number of rows in each sub-array, the fewer arrays were allocated over the time series. The more arrays allocated, the more loops were performed between Steps 2 and 5, increasing the execution time. It is, therefore, necessary to find a balance with optimum values in each sub-array, which was the purpose of the automatic selection by Equation (6) (comparisons of mean deviation with increasing sub-array size are illustrated in Appendices A and B respectively for the two case studies). The input parameters that did not maintain a positive or negative trend required more sub-arrays to account for their variation. The framework allocated the same number of sub-arrays to each input to maintain equal consideration throughout the analysis. Flexible size allocation by individual input trend or average variance rather than sample size warrants further investigation.

Step 2b defined uncertainty indicators associated with qualitative inputs. These are ideally defined by multiple sources such as surveys, interviews, and historical trends. The mean indicator is taken to calculate the Geometric Standard Deviation (GSD). Naturally, high uncertainty reflects low confidence in the measured parameter. While the use of the GSD overcame scale dependency in the measured data, the resulting coefficient of variation (CV) was found to be considerably lower than that of normally distributed data and the qualitative factors attributed by the pedigree matrix. This was due to the number of data points in the sub-array unit. Uncertainty indicators for the qualitative factors were initially assigned on a scale between 1 and 2 and the square root calculated to give the GSD [38]. These were rescaled by Equation (14) to give a more equal comparison to the quantitative data. This would, however, artificially reduce the aggregated total and saw normally distributed parameters such as $T_3$ in Case Study 1 attributing the greatest influence on the aggregated total.

Significant correlations between input variables are defined via Spearman's rank coefficient in Step 3. The ability to define the ideal coefficient limit allows the user to define the desired level of detail of the dependant variables. This can have a significant impact on the resulting estimate. The dependencies identified between the parameter values did not impact the aggregated total of the two case studies in Step 5. However, the influence attributed by individual CVs to the aggregated total in Step 6 was shown to exhibit

dependencies that warrant further investigation. Stronger dependencies between parameter values will have a greater influence on emergent behaviour in more complex systems.

The CV was adopted as the uncertainty measure in Step 4 to allow the inputs of varying distribution types to be represented on an equal scale, enabling effective uncertainty quantification. Representing uncertainty by the CV proved effective to aggregate uncertainties represented by different distributions in Step 5, but further research is required into the scaling of geometric against arithmetic standard deviations. Acceptable levels of uncertainty are user-defined according to the application and visualised by the colour scale. Conversion of further distribution types such as Weibull and non-parametric derivations will allow for the consideration of more complex datasets. Aggregating the individual CVs by a combination of the propagation of error method for symmetric CVs and the product of asymmetric CVs allowed an aggregated total estimate to be obtained. This can be used to determine how the aggregated uncertainty changes over time, which is converted back to the standard deviation and used as the response vector in Step 6.

Global Sensitivity Analysis (GSA) was employed to identify which individual uncertainties contribute the greatest influence to the aggregated total. The sampling method was applied by Groen [68] using matrix-based LCA. It was applied in this study using the individual uncertainties of each sub-array as the inputs and the aggregated uncertainty at each point as the response. It was deemed the best-suited GSA method for the CUQA framework because it can be implemented with relatively small datasets and illustrates the influence of correlated and uncorrelated uncertainties against the total effects. While the sub-array derivation in Step 2a was more accurate with a greater number of rows in each sub-array, the number of sub-arrays affected the quality of the GSA over each unit. The removal of factors that do not contribute to the aggregated total (uniformly distributed or negligible for each iteration) allowed for a focused analysis on influential parameters in a second pass through the feedback loop. The risks formed as a result of these uncertainties can then be mitigated. More in-depth GSA at each time using methods such as Sobol' indices would require the derivation of model process equations for the system application, which is out of the scope of this study.

Compared to complex engineering systems used in operational environments, Case Study 1 represented a relatively simple laboratory system setup, but served to prove the functionality of the CUQA framework as it exhibited uncertainties akin to those faced in such environments and presented comparable challenges to UQ. While the coefficients of the correlated parameters fell between negligible error margins with minimal risk, they may have a significant impact in real-world environments where operating conditions such as atmospheric temperatures or wind speeds will impact the accuracy of recorded data or subjective opinion.

The applications for complex engineering systems will feature a great deal more parameters than those exhibited in the two case studies. While the CUQA framework is able to account for additional parameters in the computation, it will take more time to produce actionable results. In addition, the visualisations resulting from Steps 2–5 would be more cumbersome to decipher with a large number of variables. This was already overcome in Step 3 by the function to only display significant correlations. The illustration of aggregated CV against individual factors for one time unit produced in Step 5 would become cumbersome with many additional parameters; however, this was only used as an example result. The use of GSA in Step 6 was even more beneficial in high-dimensional cases, where individual uncertainties that contribute the greatest influence to the aggregated total were identified.

The core contributions of the CUQA framework are:

1.    Use of the CV to enable effective quantification and aggregation of compound uncertainties represented by different distribution types;
2.    Assessment of the correlation between compound parameters;
3.    GSA for dependant compound parameters;

4. Intuitive visualisation of results showing the most-significant parameters and dominant sensitivity indices.

The authors have drawn four key conclusions as a result of this work:

1. Deriving the uncertainty measure as the CV proved effective for the aggregation of uncertainties represented by different PDFs, but further research into the scaling of geometric against arithmetic standard deviations is required. Aggregating individual CVs by a combination of the propagation of error method for symmetric CVs and the product of asymmetric CVs allowed an aggregated total estimate to be obtained. This can be used to determine how the aggregated uncertainty changes over time.

2. Dependencies between compound parameters were not found to impact the aggregated total for the two case studies. However, the influence attributed by individual CVs to the aggregated total was shown to exhibit dependencies that warrant further investigation. Such dependencies may have a significant impact in real-world environments where operating conditions such as atmospheric temperatures or wind speeds impact the accuracy of recorded data or subjective opinion.

3. The case studies served to prove the functionality of the CUQA framework, exhibiting uncertainties akin to those faced in operational environments and comparable challenges to UQ. User-defined ideal limits to identify significant correlations between compound parameters enabled the definition of the desired levels of detail for the dependant variables. Stronger dependencies between parameter values will have a greater influence on emergent behaviour in more complex systems.

4. The GSA method applied by Groen [68] was deemed the best-suited approach for the CUQA framework because it can be implemented with relatively small datasets and illustrated the influence of dependant and independent uncertainties against the aggregated total. Intuitive visualisation of the results at each stage further boosted the framework's useability and enabled rapid identification of uncertainties outside of acceptable levels and where mitigation may be required.

The authors propose future work to derive uncertainty from non-parametric and stochastic distributions through clustering techniques. Further assessment of aggregated compound uncertainty is necessary, incorporating additional distribution types and improving the rigour of the GSA approach in variance decomposition for each sub-array time unit. The emergent behaviour of uncertainties should be forecast through the in-service life to determine when and where further mitigation may be required.

**Author Contributions:** Conceptualisation, A.G., J.A.E., S.A. and Y.Z.; methodology, A.G. and J.A.E.; software, A.G.; validation, A.G., J.A.E. and S.A.; formal analysis, A.G. and S.A.; investigation, A.G. and S.A.; data curation, A.G.; writing—original draft preparation, A.G.; writing—review and editing, A.G., J.A.E., S.A. and Y.Z.; visualization, A.G.; supervision, J.A.E., S.A. and Y.Z.; project administration, A.G. and J.A.E.; funding acquisition, J.A.E. All authors have read and agreed to the published version of the manuscript.

**Funding:** This research was funded by the Engineering and Physical Sciences Research Council (EPSRC), Project Reference 1944319, and the Doctoral Training Partnership (DTP).

**Data Availability Statement:** For access to the data underlying this paper, please see the Cranfield University repository, CORD, at DOI:10.17862/cranfield.rd.13550561.

**Acknowledgments:** This research was conducted as part of a Ph.D. with collaboration between the Centre for Digital Engineering and Manufacturing (CDEM) at Cranfield University (U.K.) and BAE Systems.

**Conflicts of Interest:** The authors declare no conflict of interest.

## Appendix A. Heat Exchanger Test Rig

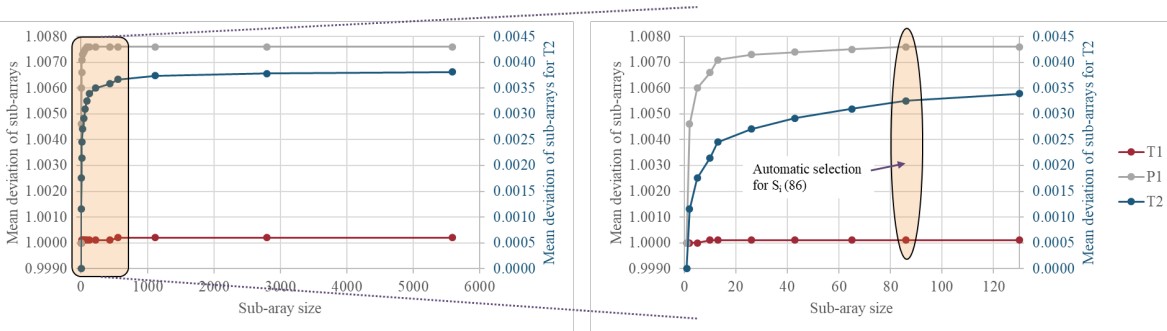

**Figure A1.** Heat exchanger test rig: increasing deviation (uncertainty) with sub-array size.

## Appendix B. C-MAPSS Turbofan Engine

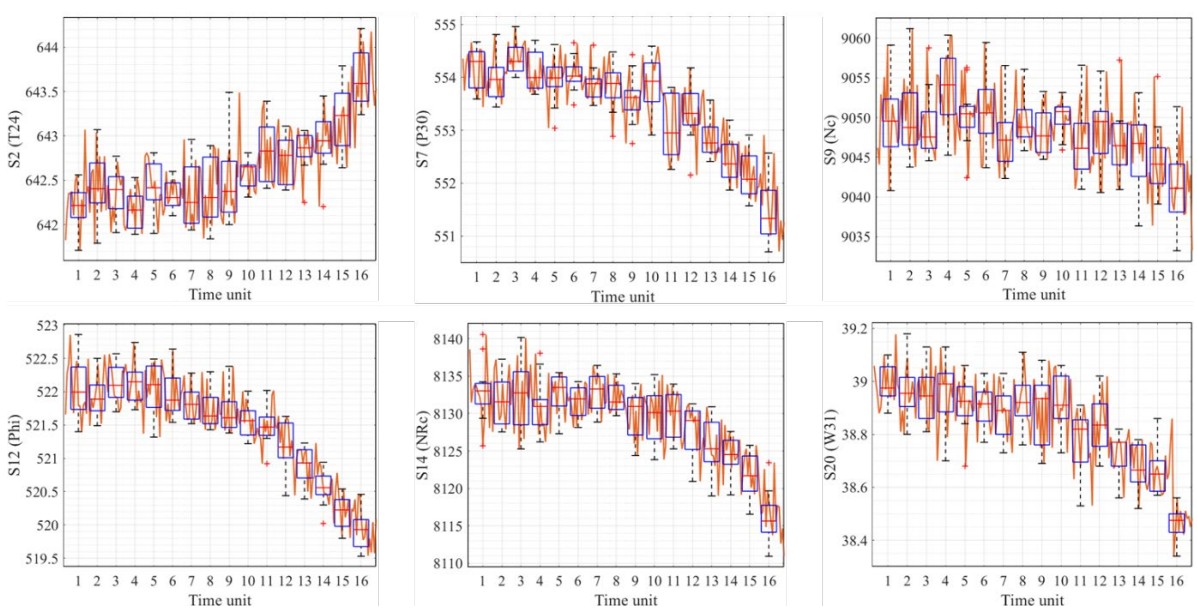

**Figure A2.** C-MAPSS turbofan engine: sub-array boxplots over time series data (example for six input parameters).

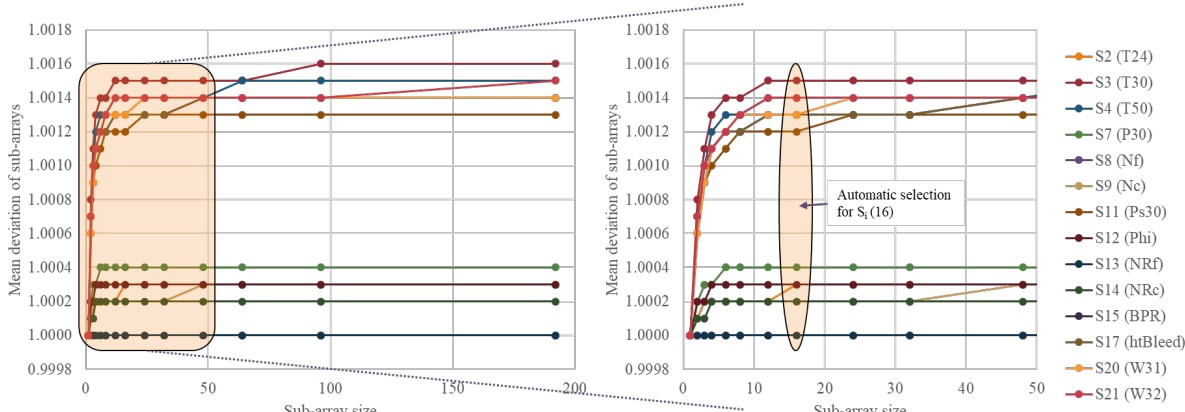

**Figure A3.** C-MAPSS turbofan engine: increasing deviation (uncertainty) with sub-array size.

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
