# Peer review of "Compound Uncertainty Quantification and Aggregation for Reliability Assessment in Industrial Maintenanceâ€"

_machines, doi:10.3390/machines11050560_

Round 1
Reviewer 1 Report (New Reviewer)

Author Response
|
Comments |
Revision |
|
1. In this paper why choose these two cases? A bespoke heat exchanger test rig and a simulated turbofan engine. Please to clarify in detail its significance, what is the relationship between the two cases? It is necessary to explain in detail.
|
The two case studies were selected to validate the developed framework. The cases were deliberately chosen for their contextual distinctness to demonstrate the framework’s flexibility in contextual application, while both exhibiting key parameter variables with compound uncertainty.
Edited in Section 3: “Here it is further developed and validated in two distinct case studies to illustrate the framework’s flexibility in contextual application, considering key parameter variables identified within a system.”
Explained at the start of Section 4.1: “The combination of digital and analogue recording, along with qualitative factors discussed below, manifests compound uncertainty in heat exchanger performance. These uncertainties need to be quantified and aggregated to assess their impact on the system, assessed via the heat transfer coefficient [25,48,77].”
Explained at the start of Section 4.2: “The C-MAPSS data consists of four datasets simulated under different operating conditions. The FD001 training dataset, simulating degradation of the high-pressure compressor (HPC), was applied to the CUQA framework to analyse the aggregated uncertainty in the measurements over time”.
|
|
2. The author mentioned the importance of Uncertainty quantification in complex system maintenance in the introduction, but did not explicitly mention it. If some engineering cases by citing other papers are added, it will be more convincing. For example, ’Review for order reduction based on proper orthogonal decomposition and outlooks of applications in mechanical systems’ is a good paper to cite.
|
Refs [1-11] in Section 1 discuss CES and UQ in various engineering fields. Section 3 states: “While maintenance practices are not discussed directly in the case studies, the compound uncertainty consideration enables greater coherence in system behaviour, reliability and maintenance requirements.”
Proper orthogonal decomposition (POD) methods are not in scope of this paper. The suggested citation is therefore not deemed relevant. It is also not openly accessible and does not mention uncertainty in the abstract or keywords. |
|
3. Some data without source description, detailed description can make the article more reasonable.
|
It is not clear what data the reviewer is referring to. |
|
4. The symbols in many formulas are not introduced in detail, which leads to the lack of rigor, such as equation 11-13 and others.
|
Description of heat load uncertainty () has been added in the text. Other parameter symbols in Eq. 11-13 are denoted previously for Eq. 10. |
|
5. The spacing in tables 1 and table 5 should be adjusted reasonably to make it easier for readers to read.
|
Table spacing is limited by the sizing and format, set under the journal manuscript template. |
|
6. The format of the references should be adjusted in time according to the requirements of the journal, such as reference 2, 3, 7…
|
All references are formatted the same way and in line with previous publications by this journal. |
|
7. English writing. There are three tenses in the abstract, general tense, past tense and future tense. Is this appropriate? |
It is generally accepted to use present tense while stating general facts (“This paper presents…”), and past tense for prior research and stating results/observations. “Results demonstrate…” has been changed to “Results demonstrated…” |
Reviewer 2 Report (New Reviewer)
- - Eq. 2 should be explained
- - Table 1. - perhaps it should be listed under the heading 2.2.1.
- - Figure 2. –this is picture containing diagram - there is no need that pictures of equipment and instruments be shown on the test rig diagram.
- - Line 406 in text – the temperature difference ???−???? should be in brackets, like this (???−????)
- - Figure 7. – numbers on the abscissa are not clearly visible. Try to find a better way to display the amount of parameters on diagram
- - Table 8. – the units for fan speed are written as Rpm and rpm. Usually it should be written as RPM or rpm. Authors are suggested tp choose one way, i.e. uppercase or lowercase, but not both. The same applies to the designation of the pressure unit (PSI or psi, but not Psi).
Author Response
Word doc attached

Reviewer 3 Report (New Reviewer)
This manuscript developed a compound uncertainty quantification and aggregation (CUQA) framework to determine compound outputs along with a determination of the greatest uncertainty contribution via global sensitivity analysis. The framework was validated with two case studies: a bespoke heat exchanger test rig and a simulated turbofan engine. Results showed the effectiveness of the framework in measuring compound uncertainty and the individual impact on system reliability. However, the following issues need to be appropriately addressed with the manuscript revised and resubmitted to the Journal for reconsideration of possible publication:
(1) As indicated in in the manuscript, this framework is an extension of the authors’ previous work, i.e., Ref. [75]. However, besides the two case studies, are there any other differences or improvements compared with the previous work [75]? Therefore, it is suggested that the authors add some additional explanations to highlight the additional work/contribution that has been achieved in this manuscript.
(2) For the two case studies, only 12 to 17 parameters/factors are considered. However, in real practices, there are quite a number of quantitative and qualitative factors to be accounted for, which may result in high-dimensional parameters (more than 50 even 100) to be considered in the proposed CUQA framework. The authors shall discuss the feasibility, including pros and cons, of applying the CUQA to high-dimensional cases.
(3) In the Literature Review, the listed literature is not up-to-date. For example:
Lines 84-85: in the listed references for aleatory uncertainty, except for [10], the other papers are not up-to-date, with the oldest one in 1997 and the latest one in 2014.
Lines 111-113: Ref. [6] and Ref. [25] were published more than 20 years ago. It is suggested to add some recent references,
Lines 184-185: The same issue happens for the literature on Sobol’ indices used in the GSA methods. It is recommended to add some up-to-date references on the GSA methods using Sobol’ indices,
(4) Some minor mistakes or typos can be found in the manuscript. The authors should carefully check the whole manuscript again. To name a few for example:
Line 92: “to consider other principals, methods …” should be “to consider other principles, methods …”;
Lines 347-348: “As discussed in Section 2.2.3, Symmetric distributions are…” should be “As discussed in Section 2.2.3, symmetric distributions are…”;
Lines 849-850: Ref. [34] is repeated with Ref. [18].
…
Author Response
Word doc attached

This manuscript is a resubmission of an earlier submission. The following is a list of the peer review reports and author responses from that submission.
Round 1
Reviewer 1 Report
Compound uncertainty quantification and aggregation (CUQA) for reliability measurement in industrial maintenance
This study presents a Compound Uncertainty Quantification and Aggregation (CUQA) framework to determine compound outputs and the most significant uncertainty contribution via global sensitivity analysis. A 6-step framework quantifies and aggregates compound uncertainties to enhance system performance assessment. Further, this was validated through two case studies: a bespoke heat exchanger test rig and a simulated turbofan engine. Results demonstrate effective measurement of compound uncertainty and the individual impact on system reliability. Overall, the manuscript is well written and can be accepted after considering the following observations
1. In the introductory section, last paragraph, page #2, the author uses the word section 0 repeatedly. Where is this section 0?
2. There are some typos in the numbering of the headings and subheadings. The author should go through it again and make any corrections.
3. The reviewer found one line as "Error! Reference source not found" throughout the manuscript. Therefore, the author should make the necessary corrections.
4. The author should improve all figures' quality (increase the text size).
Author Response
The authors thank the reviewer for their constructive feedback.
|
Comments |
Revision |
|
1. In the introductory section, last paragraph, page #2, the author uses the word section 0 repeatedly. Where is this section 0? |
The submitted manuscript used cross-referencing for headings, figures, tables and equations. The provided journal template used an outdated file type (Word 97-03). Headings and captions cannot be cross referenced. Cross references are therefore not linked, hence “Section 0” and “"Error! Reference source not found" throughout the manuscript.
The manuscript has since been typeset to correct these formatting errors.
|
|
2. There are some typos in the numbering of the headings and subheadings. The author should go through it again and make any corrections. |
Amended as for point 1. |
|
3. The reviewer found one line as "Error! Reference source not found" throughout the manuscript. Therefore, the author should make the necessary corrections. |
Amended as for point 1. |
|
4. The author should improve all figures' quality (increase the text size) |
Figure sizes have been increased. Final sizes and fitting would be determined by the journal typesetters. |
Reviewer 2 Report
The paper may be interesting but it needs significant improvement.
1. excessive references - for numerous titles it is not commented the content or the contribution of the cited paper
2. the structure of the paper is not adequate and the sections & sub-sections are numbered in a chaotic way. this makes the article to be unclear
3. in the last section, numbered 1 and entitled "Discussion and conclusions", the authors discuss several points but no factual conclusions are formulated. a section with clear conclusions is necessary to be inserted in the paper
4. the contribution of the authors should be inserted in the section "Introduction" instead of the actual position which is the section "Discussion and conclusions".
5. the "maintenance" issue found in the title of the paper is not sufficiently developed in the content. the authors should exemplify how maintenance is scheduled/influenced by the identified uncertainties. this may improve the utility of the findings of the paper
Author Response
The authors thank the reviewer for their constructive feedback.
|
Comments |
Revision |
|
1. Excessive references - for numerous titles it is not commented the content or the contribution of the cited paper.
|
The cited works are relevant to the points they are cited with. As the paper is already extensive, detailed discussion on the content and relevance of every citation was not deemed necessary. An in-depth systematic literature review on the topic has been published here: https://doi.org/10.1016/j.cirpj.2021.03.004
|
|
2. The structure of the paper is not adequate and the sections & sub-sections are numbered in a chaotic way. This makes the article to be unclear.
|
This was due to formatting errors when transferring the submitted manuscript to the journal template, which used and older file type. The manuscript has since been typeset to correct these formatting errors.
|
|
3. In the last section, numbered 1 and entitled "Discussion and conclusions", the authors discuss several points but no factual conclusions are formulated. A section with clear conclusions is necessary to be inserted in the paper.
|
Numbering has been corrected as for point 1.
4 conclusions have been listed after the contributions in the discussion and conclusions section.
|
|
4. The contribution of the authors should be inserted in the section "Introduction" instead of the actual position which is the section "Discussion and conclusions".
|
Contributions are briefly covered in the abstract (limited by word count) and in the introduction: “A 6-step framework is presented to quantify and aggregate compound uncertainties to enhance system performance assessment. This will provide maintenance planners with a comprehensive view of parameters surrounding the above factors to improve decision-making capabilities.” Detailed contributions are then covered in the final section, which is a standard format.
|
|
5. The "maintenance" issue found in the title of the paper is not sufficiently developed in the content. The authors should exemplify how maintenance is scheduled/influenced by the identified uncertainties. This may improve the utility of the findings of the paper.
|
The challenges uncertainty presents for industrial maintenance is explored in [7], as stated in the second paragraph of the introduction: “The maintenance of complex and non-complex engineering systems exhibits a range of uncertainties from interconnected factors such as quality and availability of quantitative equipment data and the qualitative influence of operators, expert opinion, experience and environmental conditions [7].” [7] is here: https://doi.org/10.1016/j.procir.2020.01.024
Updated Section 3 paragraph 1: “Every measurement or estimate is subject to a degree of error, which in turn contributes a level of uncertainty. Quantifying this uncertainty enables a thorough assessment of the scale of risk each component might inflict on the system [1,20]. The level of uncertainty and associated risk can directly or indirectly influence system reliability for maintenance planning, corresponding turnaround times and system performance.”
Updated Section 3 paragraph 3: “While maintenance practices are not discussed directly in the case studies, the compound uncertainty consideration enables greater coherence in system behaviour, reliability and maintenance requirements.” |
Reviewer 3 Report
The topic of the paper is interesting, the presentation is clear but there are very severe errors in the paper.
Eq.3 it's not clear who are xi and yi? Eq.5 it's wrong since do not consider the measurement equation.
Eq.4 is not clear, please clarify it.
Why the uncertainty is indicated with caps?
Row 213-214 pag 6 propagation of error method and GUM are two contrasting approaches, please clarify the sentence.
The reported example concerns with only multiplies between the direct measurements, but if the relationship is different equation (5) is wrong.
Author Response
The authors thank the reviewer for their constructive feedback.
|
Comments |
Revision |
|
Eq.3 it's not clear who are xi and yi? Eq.5 it's wrong since do not consider the measurement equation.
|
Added in Section 2.2.3 (denoted in Eq. 2): “(xi and yi are parameter x and parameter y where i=1 for the sum of those parameters. is the uncertainty for parameter xi).”
Eq. 5 is denoted from Muller et al [53] (https://doi.org/10.1007/s11367-014-0759-5) It is not clear what the reviewer is referring to by “the measurement equation”.
|
|
Eq.4 is not clear, please clarify it.
|
Eq. 4 was also denoted by Muller et al [53]. CVs represented by the lognormal distribution are aggregated multiplicatively.
|
|
Why the uncertainty is indicated with caps?
|
It is not clear where the reviewer is referring to. CV is in caps because it is an acronym for coefficient of variation.
|
|
Row 213-214 page 6 propagation of error method and GUM are two contrasting approaches, please clarify the sentence.
|
Updated wording to: “The GUM method is widely adopted for UQ. Along with the propagation of error method, this provides highly confident depictions of purely quantitative uncertainty. However, methods of deriving qualitative uncertainty using the GUM have been found to lead to inaccurate depictions [37,70].”
|
|
The reported example concerns with only multiplies between the direct measurements, but if the relationship is different equation (5) is wrong.
|
“Direct” measurements influence quantitative uncertainty. Both case studies consider quantitative and qualitative uncertainties. It is not clear what relationship the reviewer is referring to.
Eq. 5 is not derived by the authors. Eq. 8 denotes the method to aggregate uncertainty.
|
Reviewer 4 Report
The topic that is discussed is interesting with real world applications.
However, it is not clear what is the research beyond what was presented in the following work "An Uncertainty Quantification and Aggregation Framework for System Performance Assessment in Industrial Maintenance”.
It is strongly suggested to highlight the differences and contribution against it.
Author Response
The authors thank the reviewer for their constructive feedback.
|
Comments |
Revision |
|
1. The topic that is discussed is interesting with real world applications. However, it is not clear what the research is beyond what was presented in the paper. It is strongly suggested to highlight the differences and contribution against it.
|
Future work is proposed in the final paragraph of Section 5. This covers the research direction beyond this paper: “The authors propose future work to derive uncertainty from non-parametric and stochastic distributions through clustering techniques. Further assessment of aggregated compound uncertainty is necessary; incorporating additional distribution types and improving the rigour of the GSA approach in variance decomposition for each sub-array time unit. The emergent behaviour of uncertainties should be forecast through the in-service life to determine when and where further mitigation may be required.” |
Round 2
Reviewer 2 Report
The authors answered several questions addressed in the first review, except:
- Reducing the number of cited work, which is at this moment excessive. This is not a review paper on one hand; however, the cited papers should be commented. It is inadequate to cite "Dependencies between input parameters should be accounted for through correlation [1,8,26,39,48,57]" Are there any differences between the approaches of the authors of the articles? If the answer is YES, these should be discussed, if it is NO excessive citation is observed. Reference [52] is inserted in the reference list without containing any relevant info for the actual manuscript. For the case study 2, the authors mention that the database can be found in [79] and [80] which is incorrect. These papers just contain the source of the dataset in the reference list. Afterward, references [81] to [84] are mentioned to use this dataset, again irrelevant for a reader. Excessive and incorrect citation of sources can be observed, and a question arises: Did the authors access the dataset?
- The title is still inadequate, or the content should be adjusted. If the title contain the expression "... for reliability measurement in industrial maintenance", it should exemplify how the measured uncertainty influences the maintenance process. On the other hand, the authors do not measure uncertainty but assess it.
Thus, the answers are wrong.
The sections and sub-sections are renumbered, which permits an effective review of the paper. Some comments arise:
- The step-by-step implementation of the method should be provided, in order to give the readers' the possibility to clearly understand the approach and remake the experiments. At this stage, a sudden progress from describing an (not own) experiment to some graphics is made.
- some elements are inserted in the manuscript in a confusing order. Ex.: Table 10 is described before Table 9.
- "Random noise models were used to propagate qualitative factors associated with the simulated data with a mix of distributions to give realistic results" How? What kind of models? Is this enough to obtain reliable final results?
I consider the manuscript is not mature enough for publication. It is preferable to give a well described study case instead of 2 cases inconsistently presented.
Reviewer 3 Report
No significant improvements were made.
In my opinion the paper has to be rejected

Reviewer 4 Report
The authors present a framework for measuring uncertainty in an industrial setup with possible real-world application. The work is justified by two use cases, one of which uses simulated data, and the conclusions are in line with what the title promises.
While the work could be strengthened by adding more experiments in different industrial sectors, and discussing how the CUQA framework could be used in tandem with a production planning or similar industrial system for maintenance, i think the paper can stand as it is and the authors could consider addressing the above comments in one of their future works.
Author Response
The authors thank the reviewer for their feedback. Future work will cover applications of the CUQA framework in wider industrial settings.
Round 3
Reviewer 3 Report
The applicability of the proposal is very limited and this is not evident from the paper.
Furthermore, there are indications that do not comply with ISO GUM without real reasons.
In my opinion, the job should be rejected